# Hippocampal representations switch from errors to predictions during acquisition of predictive associations

Fraser Aitken [1,2] & Peter Kok [1✉]

We constantly exploit the statistical regularities in our environment to help guide our perception. The hippocampus has been suggested to play a pivotal role in both learning environmental statistics, as well as exploiting them to generate perceptual predictions. However, it is unclear how the hippocampus balances encoding new predictive associations with the retrieval of existing ones. Here, we present the results of two high resolution human fMRI studies (N = 24 for both experiments) directly investigating this. Participants were exposed to auditory cues that predicted the identity of an upcoming visual shape (with 75% validity). Using multivoxel decoding analysis, we find that the hippocampus initially preferentially represents unexpected shapes (i.e., those that violate the cue regularities), but later switches to representing the cue-predicted shape regardless of which was actually presented. These findings demonstrate that the hippocampus is involved both acquiring and exploiting predictive associations, and is dominated by either errors or predictions depending on whether learning is ongoing or complete.

[1] Wellcome Centre for Human Neuroimaging, UCL Queen Square Institute of Neurology, University College London, 12 Queen Square, London WC1N 3AR, UK. [2] School of Biomedical Engineering and Imaging Sciences, King's College London, St Thomas' Hospital, London SE1 7EH, UK. ✉email: p.kok@ucl.ac.uk

W e constantly exploit the statistical regularities in our environment to help guide our perception[1]. For instance, hearing a particular jingle will prime our sensory systems for the sight (and taste!) of ice cream. But how does the brain acquire and exploit knowledge about such regularities in a changing environment?

The hippocampus has been suggested to play a pivotal role in this process. That is, the hippocampus has been shown to be involved in learning novel associations between arbitrary stimuli[2–9], especially when stimuli are discontiguous in space and time[10–12], as is the case for many predictive contextual cues. In fact, learning of such relationships is strongly impaired when the hippocampus is damaged[13–18]. At the same time, the hippocampus has also been suggested to play a role in exploiting such predictive associations once learning is complete[1,19,20]. Specifically, one of the main computational functions of the hippocampus is to retrieve associated items from memory based on partial information, a process known as pattern completion[21–23]. This function has mostly been studied in the context of memory recall, but is also ideally suited for retrieving perceptual predictions based on contextual cues[5,24–28].

This raises the question of how the hippocampus balances the encoding of new associations with the retrieval of existing ones[29,30]. One way to achieve this would be to emphasise prediction errors when an environment is novel, since these can serve to update one's internal model of the world[31]. On the other hand, once an environment (and its statistical regularities) have become familiar, prediction errors may be downweighted and predictions (i.e., retrieval of existing associations) may dominate. That is, once the statistical regularities of an environment are fully learnt, the hippocampus becomes more resilient to prediction errors caused by random fluctuations (i.e., expected uncertainty), since these are no longer considered model updating ('newsworthy') events. Indeed, many previous studies have reported prediction error signals (i.e., a response evoked by a mismatch between representations retrieved from memory and current sensory inputs) in the hippocampus[32–37], while others have instead revealed prediction signals (i.e., a representation of a predicted stimulus, regardless of whether it is actually presented)[5,26,27,38]. Potentially, this seeming contradiction may arise from the fact that mismatch signals have mostly been reported in the context of episodic memory-like paradigms, where individual stimuli are only repeated a few times, whereas studies revealing prediction signals have generally involved a longer training phase to fully establish predictive associations before measuring neural signals. That is, when stimuli or associations are novel, the hippocampus is mainly driven by sensory signals that provide the opportunity to update our model of the world, i.e., prediction errors[28,39–42]. However, once learning is complete and environmental contingencies are no longer novel, hippocampal processing is dominated by retrieving predicted stimuli based on contextual cues to optimally guide perception[1,27,43,44].

In line with this idea, recent work has shown that novel prediction errors can bias the human hippocampus towards encoding[45], increasing sensory processing (i.e., EC to CA1 connectivity) and decreasing mnemonic retrieval (CA3/DG to CA1 connectivity). In addition, behavioural evidence suggests that expectation violation[46] and novelty[47] can bias the hippocampus towards performing pattern separation, proposed to underlie prediction error computations[28]. Indeed, in the context of episodic memory, it has been proposed that the hippocampus operates in two distinct modes, namely an encoding mode that prioritises processing of novel sensory signals and promotes plasticity, and a retrieval mode that prioritises memory retrieval through pattern completion[48,49]. The hippocampus is thought to be biased towards encoding by novelty-induced increases in neuromodulators such as acetylcholine (ACh) and norepinephrine (NE)[50–52] and hippocampal theta phase resets[53–55].

However, a proper test of this proposal requires establishing whether the hippocampus switches from representing errors to predictions as learning progresses. Here, we present the results of two high-resolution fMRI studies ($N = 24$ for both experiments) directly testing this hypothesis. Participants were exposed to auditory cues that predicted the identity of an upcoming visual shape (with 75% validity). To preface our findings, we found that the hippocampus initially preferentially represented unexpected shapes, but later switched to representing the cue-predicted shape regardless of whether it was actually presented. Furthermore, in this latter phase we observed increased informational connectivity between the posterior subiculum and early visual cortex (V1), in line with hippocampal predictions being relayed to the sensory cortex. These findings demonstrate that hippocampal representations switch from being dominated by errors to predictions as associative learning proceeds.

## Results

We present the results of two human fMRI studies ($N = 24$ participants in both experiments) in which human participants were exposed to auditory cues that predicted the identity of an upcoming visual shape (with 75% validity) (Fig. 1a, b). On each block of trials ($n = 32$ trials per block in Experiment 1 and $n = 128$ in Experiment 2) new auditory cues were presented, such that novel associations would have to be learnt.

Participants performed a shape discrimination task that was orthogonal to the predictive cues. Specifically, on each trial, the first shape (validly predicted, 75% of trials, or invalidly predicted, 25%) was followed by a second shape that was either identical to the first (50% of trials) or very slightly warped (50%; see "Methods" for details). Participants' task was to indicate whether the two shapes were same or different. This task was designed to encourage participants to pay attention to the shapes while keeping the cue-shape contingencies task-irrelevant. In fact, participants were not informed that the auditory cues predicted the identity of the upcoming shape, and debriefing revealed that they did not become aware of this during the experiments. In other words, any learning of cue-shape associations was incidental and implicit.

Multivoxel decoding analyses (Supplementary Fig. 1), trained on data from separate shape-only runs in which no predictive cues were presented (Fig. 1c, d), were used to reveal hippocampal shape representations on valid and invalid trials (Fig. 1e). If the hippocampus were to represent prediction errors, valid trials should not result in a shape representation, since the predicted and presented shapes are identical and should cancel each other out (Fig. 1f, top left). On invalid trials on the other hand, if shape B is predicted but shape A is presented, unexpected shape A should be represented in the hippocampus (Fig. 1f, middle left). If instead, the hippocampus was to represent predictions rather than errors, on invalid trials where shape B is predicted but shape A is presented, shape B should be represented in the hippocampus (Fig. 1f, middle right). Further, on valid trials, the shape that is both predicted and presented should be represented (Fig. 1f, top right).

Both of these types of patterns have been observed in the hippocampus[56], and the aim of the current study was to investigate how they develop over the course of learning. Note that the temporal resolution afforded by fMRI did not allow us to investigate any potential fast within-trial dynamics of these hypothesised prediction and prediction error signals. Rather, the shape representations revealed here reflect a temporal integration of neural signals over the course of a trial. It seems likely that both

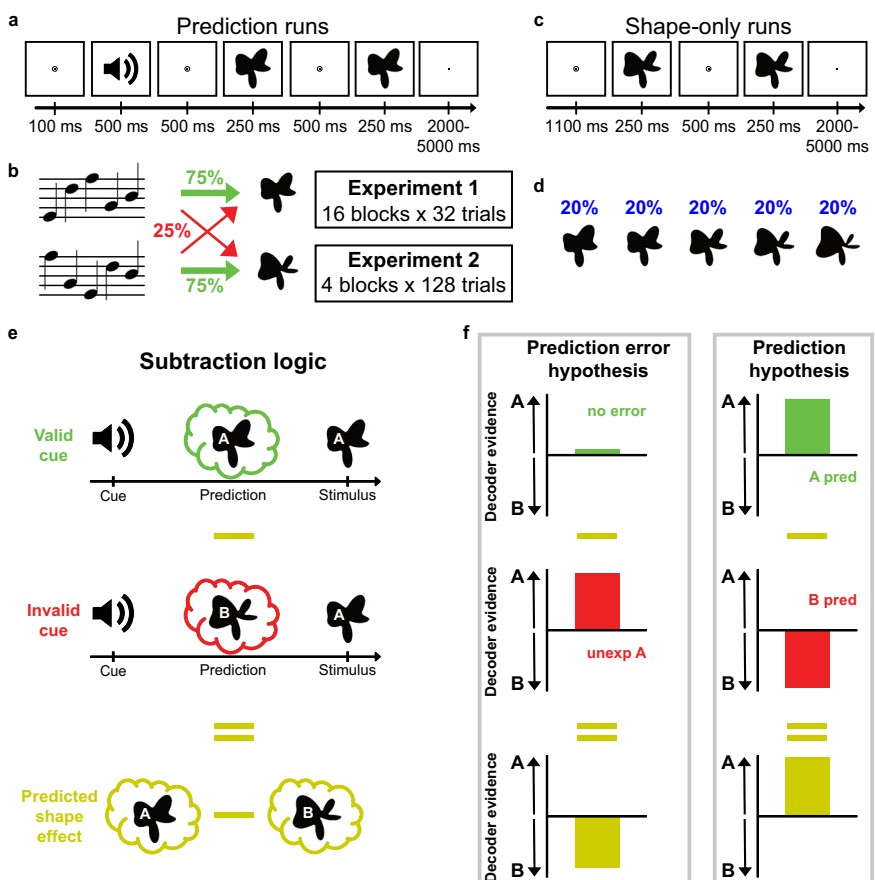

**Fig. 1 Experimental paradigm and analysis. a** During prediction runs, an auditory cue preceded the presentation of two consecutive shape stimuli. On each trial, the second shape was either identical to the first or slightly warped with respect to the first along an orthogonal dimension, and participants' task was to report whether the two shapes were the same or different. **b** The auditory cues predicted whether the first shape on a given trial would be shape 2 or shape 4 (of 5 shapes). The cue was valid on 75% of trials, whereas in the other 25% of (invalid) trials the unpredicted shape was presented. **c** During shape-only runs, no auditory cues were presented. As in the prediction runs, two shapes were presented on each trial, and participants' task was to report the same or different. **d** All five shapes appeared with equal (20%) likelihood during shape-only runs. **e** Subtracting the response evoked by invalidly from validly predicted shapes isolated the effect of the predictive cues. **f** Hypothesised shape decoding results if the hippocampus represents either prediction errors (left column) or predictions (right column).

predictions and prediction errors play a role in hippocampal computations, the question addressed here is whether the relative weighting of the two is affected by novelty and uncertainty.

The clearest way to dissociate the effects of the predictive cues from the effects of the presented shapes is to subtract decoding evidence for the invalidly predicted shapes from evidence for the validly predicted shapes (Fig. 1e), since the presented shapes were identical in both types of trials. Under a prediction error hypothesis, this would result in a negative signal (subtracting a positive signal on invalid trials from a zero signal on valid trials; Fig. 1f left column). Under a prediction hypothesis on the other hand this would result in a positive signal (subtracting a negative signal on invalid trials from a positive one on valid trials; Fig. 1f right column). This subtraction, therefore, constitutes our main effect of interest. In addition, averaging the evidence for validly and invalidly predicted shapes allowed us to quantify evidence for the shape as presented on the screen, regardless of the cues[27].

**Experiment 1: short blocks**. In Experiment 1, participants completed 16 blocks of 32 trials, with two novel auditory cues being presented in each block, while the same two visual shapes were presented throughout. Each auditory cue predicted which of the two shapes would be presented with 75% validity (Fig. 1b).

**Experiment 1—behavioural results**. Participants were able to detect small differences in the shapes, during both the shape-only runs (67.7 ± 1.7% correct; 29.7 ± 1.8% modulation of the 3.18 Hz radial frequency component, mean ± SEM) and during the prediction runs (69.0 ± 1.4% correct; 28.7 ± 1.9% modulation). Accuracy and reaction times (RTs) did not differ significantly between valid (68.9 ± 1.5% correct; RT = 592 ± 19 ms) and invalid (69.3 ± 1.6% correct; RT = 595 ± 20 ms; both $p > 0.10$) trials. Task accuracy was stable over trials and no difference between valid and invalid trials emerged over time (Supplementary Fig. 2a). This is as expected, since the discrimination task was orthogonal to the prediction manipulation (see "Methods" for details), and in line with previous results[27].

**Experiment 1—fMRI decoding results**. The dynamics of hippocampal shape representations over trials were investigated using a sliding window approach (see Methods for details). In the second half of the blocks, hippocampal activity patterns started to reflect unexpected (i.e., invalidly predicted) visual shapes (significant cluster from trial 22 to 32, $p = 0.024$; Fig. 2a, red line). However, there was no significant representation of validly predicted shapes (no clusters with $p < 0.05$; Fig. 2a, green line). In fact, there was a significant difference between invalidly and validly predicted shape decoding in the hippocampus (valid–invalid,

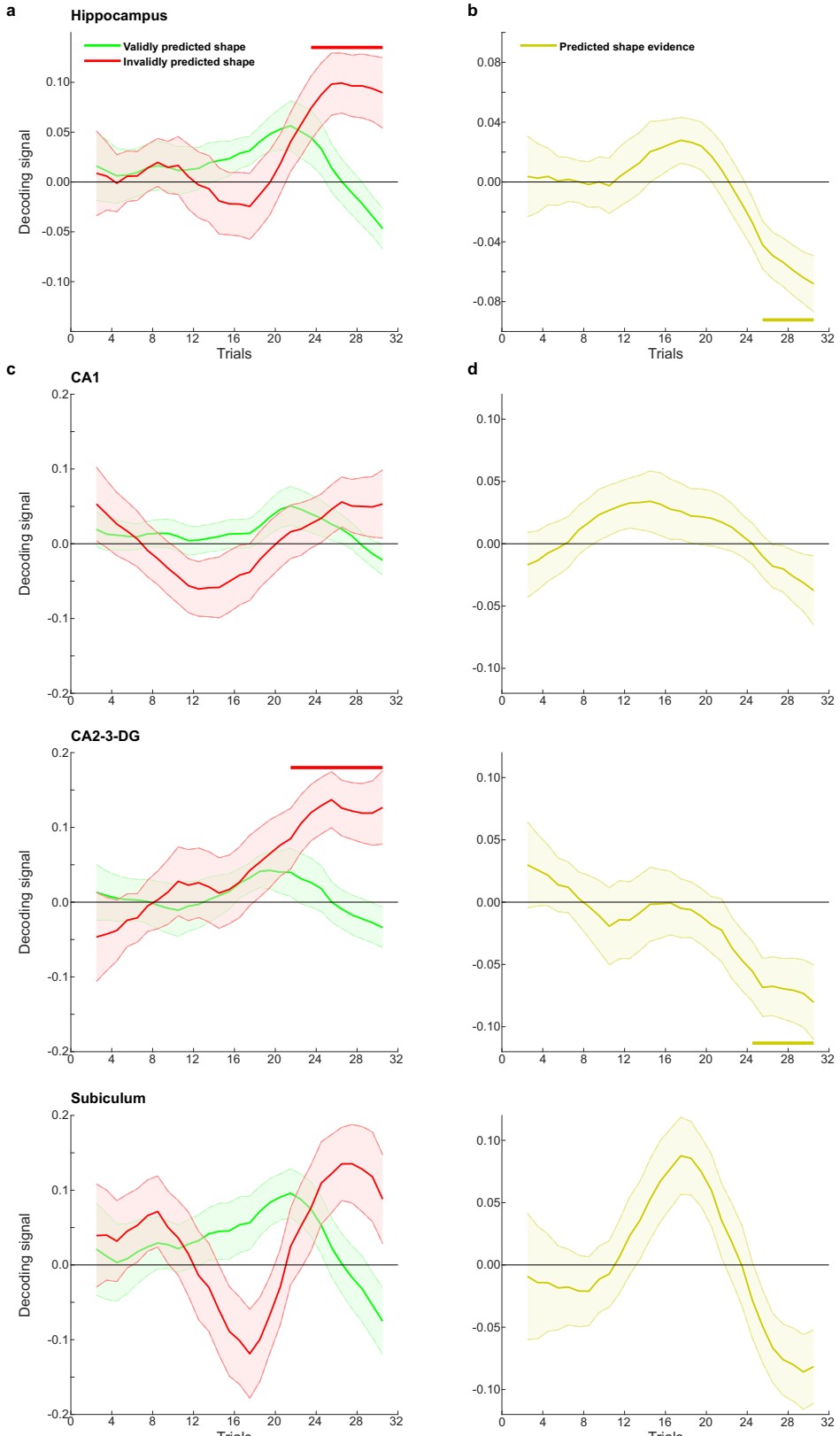

**Fig. 2 Experiment 1 shape decoding over trials. a** Decoding evidence for validly (green) and invalidly (red) predicted shapes in the hippocampus.
**b** Decoding evidence for predicted (valid–invalid) shapes in the hippocampus. **c** Decoding evidence for validly (green) and invalidly (red) predicted shapes in hippocampal subfields. **d** Decoding evidence for predicted (valid–invalid) shapes in hippocampal subfields. Time courses were temporally smoothed using a sliding window approach (see "Methods" for details). Horizontal lines indicate significant clusters. $N = 24$ participants, shaded regions indicate SEM.

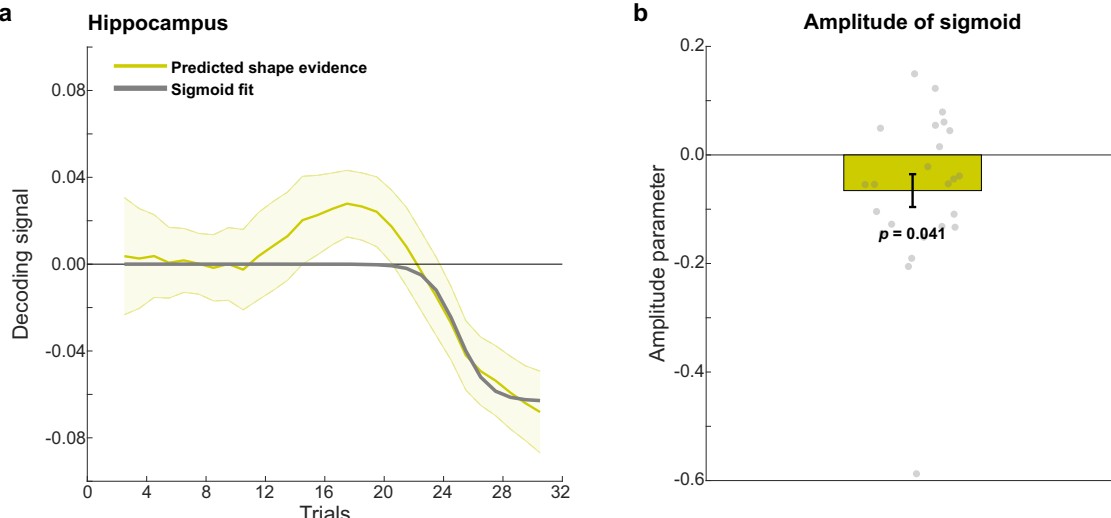

**Fig. 3 Quantification of hippocampal learning curve in Experiment 1. a** Sigmoid learning curve fit to predicted shape decoding in hippocampus. $N = 24$ participants, shaded regions indicate SEM. **b** Amplitude parameter of the sigmoid curve. $N = 24$ participants, error bars indicate SEM. $P$ value reflects two-sided one-sample $t$-test (df $= 23$) against zero. Dots indicate individual participants. Source data are provided as a Source Data file.

significant cluster from trial 24 to 32, $p = 0.028$; Fig. 2a). In other words, hippocampal activity patterns reflected shapes that were unexpectedly presented (e.g., decoding evidence for shape A when shape B was predicted but A was presented) but not shapes that were presented as expected (e.g., no evidence for shape A when shape A was predicted and presented). In sum, towards the end of the blocks, activity patterns in the hippocampus reflected a prediction error-like signal, representing unexpected but not expected shapes.

Segmenting the hippocampus into its subfields revealed that this effect was significantly present in CA2-3-DG (significant cluster for invalidly predicted shapes from trial 20 to 32, $p = 0.014$; significant cluster for valid–invalid from 23 to 32, $p = 0.045$), but not in CA1 and the subiculum (no clusters with $p < 0.05$), suggesting that this effect may have been driven by CA2-3-DG. However, decoding evidence for the predicted shape (i.e., valid–invalid, Fig. 2d) in the last bin was not significantly different between the different subfields ($F_{2,46} = 0.83$, $p = 0.44$). Given the recent interest in potential functional differences along the long axis of the hippocampus[57–59], we also compared decoding evidence for the predicted shape in the last bin between the posterior and anterior hippocampus, but found no significant difference ($t_{23} = 0.91$, $p = 0.37$). However, decoding evidence for the predicted shape was significant in the posterior ($t_{23} = -3.83$, $p = 0.00086$) but not the anterior ($t_{23} = -1.36$, $p = 0.19$) hippocampus, suggesting the posterior hippocampus may be driving the prediction error-like effects.

In order to quantify the emergence of these signals over trials, we fit sigmoid functions, or S-curves, to the decoding evidence for predicted shapes in the hippocampus (Fig. 3a; see "Methods" for details). In line with the results from the non-parametric cluster-based permutations tests reported earlier, the best fitting sigmoids had a significantly negative amplitude in the hippocampus ($t_{23} = -2.17$, $p = 0.041$) and CA2-3-DG ($t_{23} = -2.90$, $p = 0.0080$), but not in CA1 ($t_{23} = -0.31$, $p = 0.76$) and the subiculum ($t_{23} = -1.38$, $p = 0.18$). Finally, in a control analysis, to quantify the representational change over time without making any assumptions about the shape of this change, we calculated the derivative of the decoding evidence for the predicted shape over trials. In line with the previous analyses, in hippocampus ($t_{23} = -2.72$, $p = 0.012$) and CA2-3-DG ($t_{23} = -2.84$, $p = 0.0092$), but not in CA1 ($t_{23} = -0.58$, $p = 0.57$) and the subiculum

($t_{23} = -1.55$, $p = 0.13$), the average derivative over the course of the blocks was significantly negative.

It is noteworthy that, visually, an early positive predicted shape effect seemed to be present in the hippocampus (Fig. 2b), especially in the subiculum (Fig. 2d). This effect was not significant according to the cluster-based permutation tests, but in an exploratory post hoc analysis we investigated whether this early positivity was significant by fitting two sigmoids, rather than one, to the predicted shape evidence (see Methods for details). There was no significantly positive early sigmoid in hippocampus as a whole ($t_{23} = 1.13$, $p = 0.27$), nor in CA1 ($t_{23} = 0.77$, $p = 0.45$) or CA2-3-DG ($t_{23} = 0.66$, $p = 0.52$), but there was in the subiculum ($t_{23} = 3.43$, $p = 0.002$; Supplementary Fig. 3).

Based on previous findings of predictive signals in the caudate nucleus[4,27,60], we also tested these effects in the caudate, and found that like the hippocampus, caudate activity patterns reflected unexpected (significant cluster from trial 20 to 29, $p = 0.0062$) but not expected (no clusters with $p < 0.05$) shapes towards the end of the blocks, with a significant difference between the two conditions (valid–invalid, significant cluster from trial 20 to 29, $p = 0.033$; Supplementary Fig. 4).

The fact that the hippocampus displayed a prediction error-like pattern (cf. Figs. 2a and 1f, left column) is striking given that several previous studies have reported prediction-like effects[5,26,38]. Specifically, a previous study with a virtually identical design[27] revealed evidence for the shape predicted by the cue, regardless of which shape was actually presented (as in Fig. 1f, right column). The crucial difference is that in these previous studies participants were exposed to the predictive associations for many trials before the fMRI session, whereas here participants learned novel predictive associations every block. Based on this, we hypothesised that the hippocampus may switch from representing prediction errors (early in learning) to representing predictions (once learning is complete) as learning progresses (Fig. 4). In order to test this hypothesis, we performed a second fMRI experiment, in which participants ($N = 24$) were exposed to the same cues for longer, and tested for potential switches in dynamics by fitting sigmoid learning curves to the decoding evidence over trials.

**Experiment 2: long blocks.** In Experiment 2, participants were exposed to 4 blocks of 128 trials (compared to 16 blocks of 32

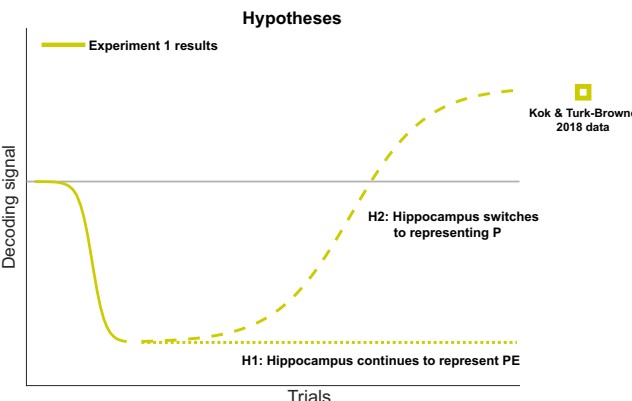

**Fig. 4 Hypotheses for Experiment 2.** Experiment 1 revealed a negative shape decoding signal (solid line). Lengthening the learning phase may either result in this effect continuing (H1, dotted line), or lead to a switch towards positive prediction signals once learning is complete (H2, dashed line). Square indicates results from ref. [27], where participants were acquainted with the predictive cues before the fMRI session.

trials in Experiment 1), with new auditory predictive cues being presented in each block. In all other regards, Experiment 2 was identical to Experiment 1.

**Experiment 2—behavioural results.** As in Experiment 1, participants were able to detect small differences in the shapes, during both the shape-only runs (69.5 ± 1.7% correct; 30.1 ± 2.0% modulation of the 3.18 Hz radial frequency component, mean ± SEM) and during the prediction runs (68.5 ± 1.8% correct; 24.5 ± 2.0% shape modulation). Accuracy and reaction times (RTs) again did not differ significantly between valid (68.6 ± 1.9% correct; RT = 651 ± 18 ms) and invalid (68.2 ± 1.9% correct; RT = 654 ± 18 ms; both $p > 0.10$) trials. Task accuracy was stable over trials and no difference between valid and invalid trials emerged over time (Supplementary Fig. 2b).

**Experiment 2—fMRI decoding results.** As in Experiment 1, dynamics of hippocampal shape representations over trials were investigated using a sliding window approach (see Methods for details). Reflections of validly and invalidly predicted shapes in hippocampal activity patterns displayed striking of dynamics over time (Fig. 5a). These dynamics were quantified by fitting two sigmoid curves to the decoding evidence for predicted shapes (Fig. 5b), one with an inflection point in the first half of the blocks (trials 1–64) and the other with an inflection point in the second half (trials 65–128). This analysis revealed that an initial negative curve (amplitude parameter of early sigmoid; $t_{23} = -2.26$, $p = 0.033$), reflecting evidence for unexpected but not expected shapes (i.e., prediction error, as in Experiment 1) was followed by a positive curve (amplitude parameter of the late sigmoid curve; $t_{23} = 2.45$, $p = 0.022$) about halfway through the blocks. This switch was most striking for invalidly predicted shapes (Supplementary Fig. 5a). Initially, approximately halfway through the blocks, the unexpectedly presented shape was represented (Supplementary Fig. 5b). However, at the end of the blocks, the hippocampus instead represented the shape predicted by the auditory cue, rather than the shape presented on the screen (Supplementary Fig. 5b).

Segmenting the hippocampus into its subfields revealed that both the early negative and later positive learning curves were also significant in CA1 (early: $t_{23} = -2.75$, $p = 0.011$; late: $t_{23} = 2.96$, $p = 0.0070$), but not in CA2-3-DG (early: $t_{23} = -0.95$, $p = 0.35$; late: $t_{23} = 0.54$, $p = 0.60$), while in the subiculum the early negative

curve was marginal ($t_{23} = -2.07$, $p = 0.05$) while the later positive one was significant ($t_{23} = 2.56$, $p = 0.018$).

As in Experiment 1, we performed a control analysis that did not make any assumptions about the shapes of the learning curves, in which we calculated the derivative of the decoding evidence for the predicted shape (Fig. 5b, e), separately for the first and second half of the blocks. In line with the curve fitting results, the derivative was significantly different in the first versus the second halves of the blocks in hippocampus ($t_{23} = -2.67$, $p = 0.014$) and CA1 ($t_{23} = -2.41$, $p = 0.024$), while this difference was marginal in CA2-3-DG ($t_{23} = -2.06$, $p = 0.051$) and not significant in the subiculum ($t_{23} = -1.34$, $p = 0.19$). This was driven by the derivative being significantly positive in the second half of the blocks in hippocampus ($t_{23} = 2.25$, $p = 0.034$) and CA1 ($t_{23} = 2.24$, $p = 0.035$), but marginally negative in the first half (hippocampus: $t_{23} = -2.00$, $p = 0.057$; CA1: $t_{23} = -1.98$, $p = 0.060$). In CA2-3-DG, the derivative was significantly negative in the first half ($t_{23} = -2.73$, $p = 0.012$) but not the second half ($t_{23} = 0.87$, $p = 0.39$), while neither half was significant in the subiculum (first half: $t_{23} = -0.51$, $p = 0.61$; second half: $t_{23} = 1.49$, $p = 0.15$). There was no significant difference between the hippocampal subfields in terms of the derivative of the decoding time courses in either the first ($F_{2,46} = 0.58$, $p = 0.56$) or second halves ($F_{2,46} = 0.79$, $p = 0.46$) of the blocks.

However, there was a significant difference between posterior and anterior hippocampus, with the positive derivative in the second half of the blocks being stronger in posterior than anterior hippocampus ($t_{23} = 3.00$, $p = 0.0064$; Supplementary Fig. 6). In fact, the early negative (posterior: $t_{23} = -2.43$, $p = 0.024$; anterior: $t_{23} = -0.86$, $p = 0.40$) and late positive (posterior: $t_{23} = 2.70$, $p = 0.013$; anterior: $t_{23} = 0.98$, $p = 0.34$) sigmoids, as well as the difference in the derivative between the first and second halve of the blocks (posterior: $t_{23} = 2.74$, $p = 0.012$; anterior: $t_{23} = 1.42$, $p = 0.17$), were significant in the posterior, but not anterior hippocampus.

A positive slope in the second half of the blocks might also be observed if the early prediction error signal simply gradually disappeared, rather than hippocampal representations switching to a positive prediction signal. To resolve this, we tested whether decoding evidence for the predicted shape at the end of the blocks (i.e., the final bin) was significantly larger than zero. While this signal was not significantly positive for the hippocampus as a whole ($t_{23} = 1.75$, $p = 0.094$), it was in the posterior hippocampus ($t_{23} = 2.15$, $p = 0.042$), which was also the driver of the results reported above. In fact, decoding evidence for the predicted shape at the end of the blocks was stronger in the posterior than the anterior ($t_{23} = 0.31$, $p = 0.76$; posterior vs. anterior: $t_{23} = 2.14$, $p = 0.043$) hippocampus. The significant effect in posterior hippocampus was reflected by significant evidence for the predicted shape in the posterior subiculum ($t_{23} = 2.55$, $p = 0.018$) and CA1 ($t_{23} = 2.17$, $p = 0.041$), but not CA2-3-DG ($t_{23} = 1.31$, $p = 0.20$). There was no significant evidence for the presented shape (quantified by averaging evidence for valid and invalid shapes, thereby averaging out the effect of the cues[27]; Fig. 1e) at the end of the blocks in the hippocampus ($t_{23} = -1.30$, $p = 0.21$) or any of its subdivisions (all $p > 0.2$). In other words, once predictions were learnt, hippocampal representations were determined by the predictive cues, not by which shape was actually presented on screen. Note that this analysis was based on a relatively small subset of trials (i.e., only those in the last bin of each block), but the positive evidence for predicted shapes and absence of evidence for the presented shapes is in line with a study employing a virtually identical paradigm where participants learnt the cue-shape predictions before being tested in the scanner[27]. There was no significant difference in predicted shape

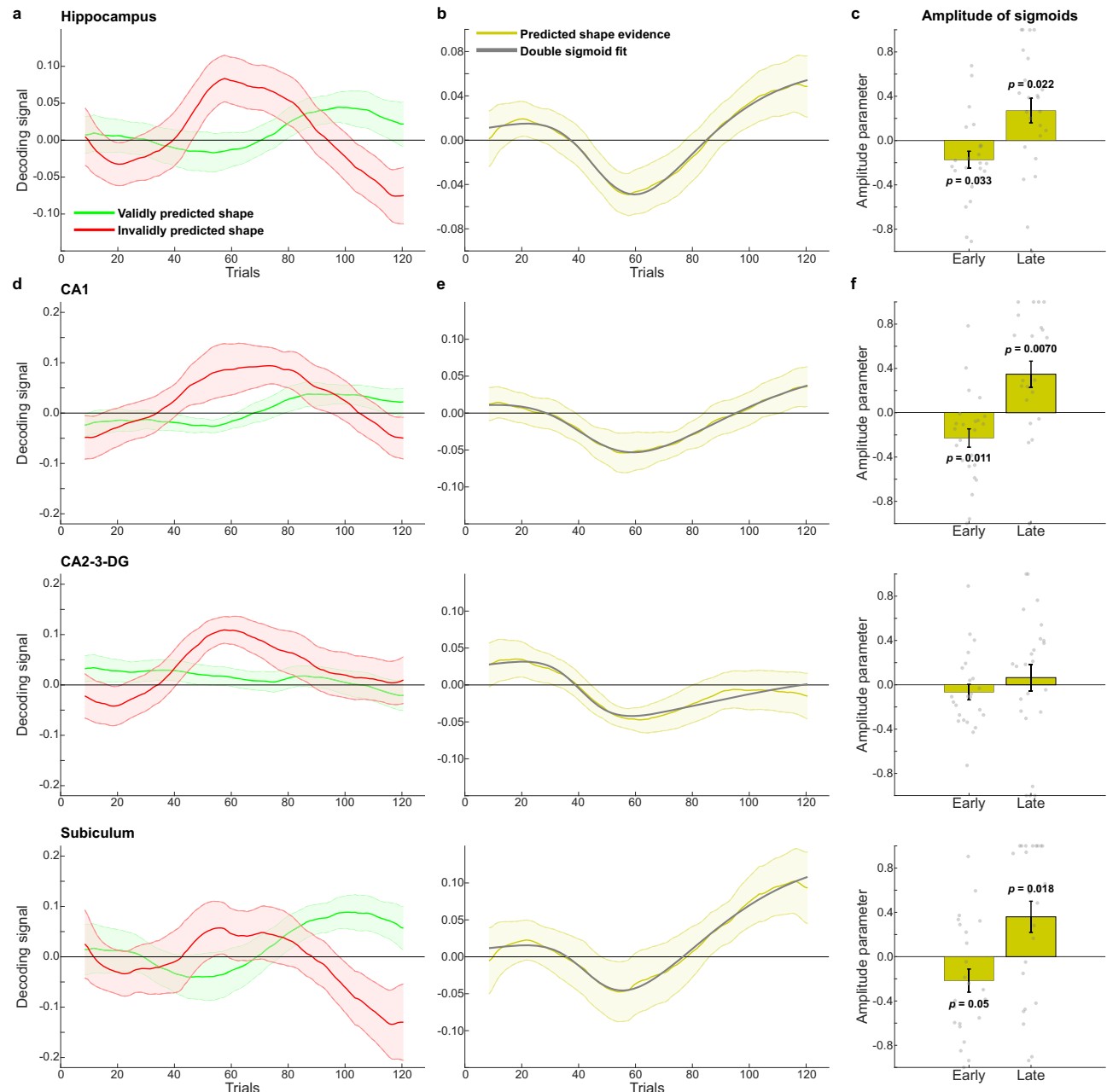

**Fig. 5 Experiment 2 shape decoding over trials. a** Decoding evidence for validly (green) and invalidly (red) predicted shapes in the hippocampus.
**b** Decoding evidence for predicted (valid–invalid) shapes in the hippocampus (yellow) with the double sigmoid fit (grey). **c** Amplitude parameters of early
(midpoint between trials 1 and 64) and late (midpoint between trials 65 and 128) sigmoid curves in the hippocampus. $P$ value reflects two-sided one-
sample $t$-test against zero. **d** Decoding evidence for validly (green) and invalidly (red) predicted shapes in hippocampal subfields. **e** Decoding evidence for
predicted (valid–invalid) shapes in hippocampal subfields (yellow) with the double sigmoid fit (grey). **f** Amplitude parameters of early (midpoint between
trials 1 and 64) and late (midpoint between trials 65 and 128) sigmoid curves in hippocampal subfields. Time courses were temporally smoothed using a
sliding window approach (see Methods for details). $N = 24$ participants in all panels. Shaded regions and error bars indicate SEM. Dots indicate individual
participants. $P$ values reflect two-sided one-sample $t$-tests (df = 23) against zero. Source data are provided as a Source Data file.

evidence between the posterior subfields ($F_{2,46} = 1.09$, $p = 0.34$),
but it is worth noting that the effect was numerically largest in the
posterior subiculum (0.13; Supplementary Fig. 7), in line with the
previous work[27].

Since the subiculum is a major output relay from the
hippocampus to neocortex[61,62], we speculate that this may be
in line with hippocampal prediction signals being communicated
to the sensory cortex, in order to guide perception. If this were the
case, one would expect functional connectivity between hippo-
campus and neocortex to increase as predictive associations are

established[63,64]. We tested this hypothesis in an exploratory
analysis of informational connectivity between the posterior
subiculum and entorhinal cortex (EC; a major interface between
hippocampus and cortex) as well as the visual cortex (V1, V2 and
LO) (see "Methods" for details). This analysis revealed increased
informational connectivity at the end of the blocks (final bin)
versus at the start of the blocks (first bin) between the posterior
subiculum and EC ($t_{23} = 2.40$, $p = 0.025$; Fig. 6) and V1
($t_{23} = 2.88$, $p = 0.0084$), but not V2 ($t_{23} = 1.73$, $p = 0.097$) and
LO ($t_{23} = -0.28$, $p = 0.78$).

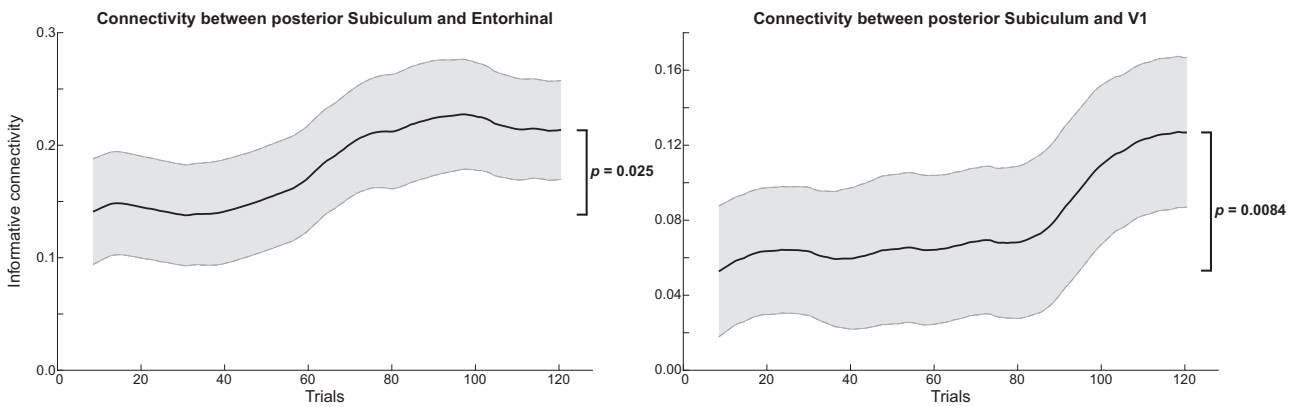

**Fig. 6 Experiment 2 informational connectivity between posterior subiculum and neocortex over trials.** Time-resolved Pearson correlation between shape evidence in the posterior subiculum and entorhinal cortex (left panel) and V1 (right panel). Time courses were temporally smoothed using a sliding window approach (see Methods for details). $N = 24$ participants, shaded regions indicate SEM. P values reflect two-sided one-sample $t$-tests (df = 23) against zero.

The caudate nucleus displayed a qualitatively similar pattern of results as the hippocampus, with an initial negative sigmoid curve ($t_{23} = -3.11$, $p = 0.0049$) being followed by a positive curve ($t_{23} = 3.65$, $p = 0.0013$; Supplementary Fig. 8a). The control analysis also revealed a significant difference in derivative in the first versus second half of the blocks ($t_{23} = 2.90$, $p = 0.0081$), driven by positive derivative in the second half ($t_{23} = 3.94$, $p = 0.00066$) but not the first half ($t_{23} = -0.33$, $p = 0.74$), and a significantly positive representation of the predicted shape in the final bin of the blocks ($t_{23} = 3.06$, $p = 0.0055$).

To investigate the specificity of these effects, we also analysed regions in the early visual cortex (Supplementary Fig. 8b–d) and the amygdala (Supplementary Fig. 8e), a region adjacent to the hippocampus. These regions did not significantly reflect the learning of predictive associations (no significantly negative or positive sigmoid curves, all $p > 0.05$), indicating that the effects in the hippocampus and caudate were region-specific[27].

## Discussion

In two human fMRI experiments, we find that as learning of associative predictions progresses, the hippocampus switches from preferentially representing unexpected stimuli (i.e., prediction errors) to representing predicted shapes. These findings demonstrate that the hippocampus is involved in both acquiring and exploiting predictive associations, and is dominated by either errors or predictions depending on whether learning is ongoing (i.e., when prediction errors are informative[28]) or complete (only expected uncertainty remains). Concretely, what this suggests in the context of the current study is that prediction errors caused by early cue violations, when learning is still very much ongoing, dominate processing in the hippocampus, leading to the representation of the unexpectedly presented shape. On the other hand, once the 75–25% cue contingencies are firmly learnt, the 25% cue violations are no longer treated as model updating ('newsworthy') events[28], are therefore no longer upweighted, and the retrieval of the cued shape dominates. Note that we are not suggesting that this is an all-or-nothing switch; it is likely that the hippocampus always represents both predictions (through pattern completion in CA3[21,24,65,66]) and errors (potentially through mismatch comparison in CA1[34,36,67]), but that the balance between the two depends on contextual factors such as novelty and unexpected uncertainty. An analogous switch between prediction vs. surprise dominated representations has recently been proposed in the realm of perception, albeit on a sub-second time-scale[68].

The early bias towards prediction errors is in line with recent demonstrations of hippocampal mode switches induced by novel prediction errors in humans[45]. Mechanistically, this switch may occur since novelty leads to an increase of neuromodulators like ACh and NE[50–52], which suppress retrieval-related connections (CA3's autorecurrence and CA3 -> CA1) relative to encoding-related ones (EC -> CA1)[69–71]. Alternatively, novelty may promote encoding on a faster time-scale by inducing a hippocampal theta phase reset[49,53–55]. Further research is needed to determine whether the switch demonstrated here was indeed driven by hippocampal mode changes or by a different mechanism that upweights novel prediction errors, such as attention[72–74]. For instance, methods with higher temporal resolution such as EEG/MEG or invasive electrophysiology could be used to investigate whether there is a relationship between hippocampal theta phase and error vs. prediction representations in the hippocampus. In either case, as learning progresses and novelty diminishes, a bias towards encoding prediction errors is abolished and retrieval of predictive associations dominates.

As prediction signals emerged in the hippocampus, functional connectivity increased between the posterior subiculum and the entorhinal cortex and the primary visual cortex, demonstrating a potential route for relaying predictions to the sensory cortex[26,62,63,75,76]. This relaying of predictions likely involves the same mechanisms that are responsible for hippocampus-mediated cortical reinstatement of memories[77–80]. Of course, fMRI connectivity analyses cannot determine directionality given the slow nature of the BOLD signal, so future research using electrophysiology[81,82] or layer-specific fMRI[76,83,84] will be required to test this hypothesis further. Similarly, the slow nature of the BOLD signal prevents investigating fast within-trial dynamics of prediction and prediction error signals. For instance, it may be that prediction signals always precede prediction error signals in the hippocampus. The hippocampal representations revealed here reflect a temporal integration of neural signals over the course of a trial, and thus indicate whether predictions or prediction errors dominate. Future studies with millisecond temporal resolution are required[9,82] to reveal the dynamic interplay of predictions and errors within the hippocampus.

An exploratory post hoc analysis of Experiment 1 additionally revealed early prediction-like signals in the subiculum, before the prediction error-dominated signals emerged (Supplementary Fig. 3). This initial positive signal could potentially reflect early, imprecise predictions, which lead to strong prediction errors on subsequent invalid trials. This explanation is currently speculative, especially given the post hoc nature of the analysis. Future

research is needed to investigate the early build-up of prediction signals further, for instance using a paradigm with many blocks with only a few cue repetitions each.

It appears that the prediction error-like signals emerged later in Experiment 2 than in Experiment 1 (cf. Figs. 2 and 5). This is likely the result of differences in sliding window length and smoothing between the two experiments, combined with the fact that the prediction error signal may still be building by the end of the blocks in Experiment 1 (Fig. 2b). More speculatively, the later negative peak in Experiment 2 may also partly have resulted from averaging together the small early positive peak discussed above (~trial 16–20) with the subsequent negative signal (~trial 24 onwards) (Supplementary Fig. 3). While the sliding window in Experiment 1 was short enough to separately resolve these two signals, the longer sliding window in Experiment 2 was not, resulting in these two signals cancelling each other out early in the blocks. At present this explanation is highly speculative since the initial positive signal in Experiment 1 was detected using post hoc analyses and needs to be investigated further, as discussed above.

It is noteworthy that the predictive associations studied here were fully implicit. Participants were not informed that there were any such associations, the predictions were incidental to the task, and debriefing indicated that participants did not become aware of them over the course of the experiments[8,44]. The fact that such implicit associations still involved the hippocampus is in line with theories of hippocampal processing based on the types of computations required, rather than whether they are explicit or implicit[23,40,85]. In fact, it has even been suggested that the hippocampus may engage in error-driven conjunction learning specifically when associations are incidental to the task participants perform[86].

Despite the predictive associations being implicit, hippocampal signals may still have been affected by fluctuations in the level of attention paid to the cues over the course of the blocks. That is, if participants pay more (less) attention to the cues over time, this might increase (decrease) the strength of the prediction signals in the hippocampus. Future research might dissociate learning dynamics and attentional fluctuations by changing the reliability of predictive cues between blocks. More reliable (e.g., 90% valid) cues would be expected to lead to faster learning rates than less reliable (e.g., 60% valid) ones, without affecting non-specific fluctuations in attention due to time spent on task.

In the current study, both prediction error and prediction signals seem to have been driven by the posterior rather than the anterior hippocampus. This finding is in line with suggestions that hippocampal representations increase in complexity and scale along the long axis[57]; simple cue-stimulus associations as studied here may therefore be encoded in the posterior hippocampus[58], whereas more complex representations such as narratives[57] and scenes[87–89] are encoded in the anterior hippocampus.

Analysis of the caudate nucleus revealed similar prediction signals as in the hippocampus, in line with previous work employing a highly similar experimental design[27], as well as other studies revealing the involvement of the caudate in predictive processing[4,60]. Recently, it has been suggested that perceptual expectation signals in the tail of the striatum play a role in generating hallucination-like percepts in mice[90]. Future research is needed to establish whether the caudate and hippocampus play different or complementary roles in the processing of predictive associations[91,92].

In the current study, novel predictive cues were introduced on each block of the experiment. It is an open question whether similar hippocampal dynamics would occur if the cue identities remained the same throughout the experiment, but the predictive contingencies switched. In other words, does the hippocampal switch observed here depend on the cues themselves being novel,

or is it sufficient for only their predictive values to change, i.e., for there to be unexpected uncertainty[52]?

In addition, whether the hippocampus signals predictions or prediction errors may also depend on the type of predicted stimulus. For instance, in previous work, we reported hippocampal prediction signals for complex shapes, but prediction error-like signals for low-level features, i.e., the predicted orientation of a grating stimulus[56]. Future work systematically manipulating the complexity of visual stimuli may shed light on this by exploring the relationship between hippocampal computations and stimulus complexity[93].

In sum, the current findings demonstrate a role for the hippocampus in both acquiring and exploiting predictive associations, bridging the fields of learning and perception. These fields have separately made progress in investigating the roles of prediction, novelty and uncertainty[1,52], but have until now largely remained segregated literatures, despite great promise to inform one another[68,94]. Ultimately, weighting predictions and errors according to their reliability is crucial to optimally perceive and engage with our environment, and the current findings suggest that the hippocampus plays a crucial role in this process.

## Methods

**Participants**. Both experiments aimed to recruit 24 healthy, right-handed, MR-compatible participants with normal or corrected-to-normal vision. All participants provided informed consent through a protocol reviewed by the University College London (UCL) Research Ethics Committee and were compensated a total of £27.50 for their time. Twenty-nine individuals completed Experiment 1, of which five were excluded due to our strict head motion criteria (five or more movements larger than 1.5 mm in any direction between successive functional volumes). The final sample consisted of 24 participants (12 female; age 25.6 ± 7.2, mean ± SD). Twenty-nine individuals completed Experiment 2, of which two were excluded for not performing the task above chance, and three due to excessive head motion (see criteria above). The final sample consisted of 24 participants (19 female; age 26.2 ± 7.0, mean ± SD).

**Stimuli**. Visual and auditory stimuli were generated using MATLAB (Mathworks, Natick, MA, USA) and the Psychophysics Toolbox[95]. In the MR scanner, visual stimuli were displayed on a rear projection screen using a projector (1600 × 1200 resolution, 60 Hz refresh rate) against a grey background. Participants viewed the visual display through a mirror that was mounted on the head coil. The visual stimuli consisted of complex shapes defined by radial frequency components (RFCs)[96,97], identical to the shapes used in Kok & Turk-Browne[27] (Fig. 1). The contours of the stimuli were defined by seven RFCs, and a one-dimensional shape space was created by varying the amplitude of three out of the seven RFCs[27]. Specifically, the amplitudes of the 1.11, 1.54 and 4.94 Hz components increased together, ranging from 0 to 36 (first two components), and from 15.58 to 33.58 (third component). Note that we chose to vary three RFCs simultaneously, rather than one, to increase the perceptual (and neural) discriminability of the shapes. Five shapes (Fig. 1d) were selected from this continuum such that they represented a perceptually symmetrical sample of this shape space (see Kok & Turk-Browne[27] for details). In addition, a fourth RFC (the 3.18 Hz component) was used to create slightly warped versions of the five shapes, to enable the same/different shape discrimination cover task (see below). Experiments 1 and 2 presented identical shapes (black, subtending 4.5°), centred on fixation.

In the scanner, auditory stimuli were presented using MR-compatible ear buds (E-A-RTONE 3 A, 10 Ohm, Etymotic Research, Elk Grove Village, IL, USA). The auditory stimuli consisted of sequences of pure tones, ranging in frequency from 261.36 Hz (C$_4$) to 987.77 (B$_5$) Hz (set of 14 tones: C$_4$, D$_4$, E$_4$, F$_4$, G$_4$, A$_4$, B$_4$, C$_5$, D$_5$, E$_5$, F$_5$, G$_5$, A$_5$, B$_5$; duration = 100 ms; 10 ms linear rise and fall ramps). Seventeen sequences of five tones (500 ms) were created by selecting the least correlated sequences from all permutations of {1, 2, 3, 4, 5}. These 17 sequences (e.g., 1-2-3-4-5, 1-5-4-3-2, 3-1-5-4-2, etc.) were further differentiated by assigning them different starting tones, in the step of 3. For instance, if sequence 1 was 1-2-3-4-5, sequence 2 was 4-8-7-6-5, sequence 3 was 9-7-11-10-8, etc. Since the maximum starting tone was 10, given the set of 14 tones, every fifth sequence started with starting tone 1 again. For each sequence, a mirrored sequence was generated in order to create 17 pairs of easily distinguishable sequences consisting of the same tones (e.g., sequence 1-2-3-4-5 was paired with 5-4-3-2-1, sequence 4-8-7-6-5 was paired with 8-4-5-6-7, etc.). In Experiment 1, 16 of these pairs were assigned in random order to the sixteen blocks, while the seventeenth pair was used during the practice block outside the scanner (see below). In Experiment 2, only the first four pairs were used, since this experiment contained only four blocks (see below), while the fifth pair was used for practice.

**Experimental procedure**. The trial structure was identical in both experiments. The start of each trial was signalled by the presentation of a fixation bullseye (diameter, 0.7°). During prediction runs, an auditory cue (sequence of five tones; 500 ms) was presented 100 ms after the trial onset (Fig. 1a). Following a 500 ms delay, two consecutive shapes were presented for 250 ms each, separated by a 500 ms fixation screen. The auditory cues predicted whether the first shape on that trial would be shape 2 or shape 4 (out of five shapes; Fig. 1b, d). The cue was valid in 75% of trials, whereas in the other 25% of trials the unpredicted shape would be presented. For instance, a specific auditory cue might be followed by shape 2 in 75% of trials and by shape 4 in the remaining 25% of trials. On each trial, the second shape either was identical to the first (50%), or slightly warped (50%), by modulating the amplitude of the 3.18 Hz RFC component defining the shape. This modulation could be either positive or negative (counterbalanced over conditions) and the participants' task was to indicate whether the two shapes on a given trial were the same or different, using an MR-compatible button box (750 ms response interval). This task was designed to encourage participants to attend the visual shapes, while avoiding a relationship between the perceptual prediction and the task response. Furthermore, by modulating one of the RFCs that was not used to define our one-dimensional shape space, we ensured that the shape change on which the task was performed was orthogonal to the changes that defined the shape space, and thus orthogonal to the shape features predicted by the auditory cues. The size of the shape modulation was determined by a staircasing procedure[98], updated after every trial to ensure sufficient task difficulty (~75% correct). The end of each trial was signalled by replacing the fixation bullseye with a single fixation dot, encouraging participants to continue to fixate (inter trial interval jittered between 1.25 and 4.25 s).

Experiment 1 consisted of 16 blocks of 32 trials, presented in four prediction runs (4 blocks per run, 30 s breaks between runs, ~12 min per run). In each block, a different pair of cues were presented. For each trial number (1–32) we counterbalanced (1) which cue was presented, (2) whether the cue was valid (75%) or invalid (25%), and (3) whether the two shapes were the same or different.

Experiment 2 consisted of 4 blocks of 128 trials (1 block per prediction run, 30 s break halfway, ~12 min per run), with a different pair of cues presented in each block. As in Experiment 1, cue validity was counterbalanced for every trial position, but given the smaller number of blocks, the presented cue and shape modulation were counterbalanced over groups of four trial positions (trials 1–4, 5–8, etc.) rather than for every trial position. This was reflected in the analyses by a fourfold increase in the trial averaging window; see below.

In both experiments, which pair of cues was assigned to which block, as well as which member of each pair predicted which shape, was counterbalanced across participants.

In addition to the four prediction runs, both experiments also contained two shape-only runs, flanking the prediction runs, constituting the first and last (sixth) runs of the experiments. In these runs (120 trials per run, ~12 min) no auditory cues were presented (Fig. 1c). As in the prediction runs, each trial started with the appearance of a fixation bullseye followed 1100 ms later by two shapes (250 ms each, 500 ms interval). On each trial, one of the five possible shapes was presented, with equal (20%) likelihood (Fig. 1d). As in the prediction runs, the participants' task was to indicate whether the two shapes were the same or different. The size of the shape modulations was controlled by a staircase separate from that of the prediction runs, to equate task difficulty in these runs with five instead of two possible initial shapes. The shape-only runs acted as the training data for our shape decoding model, see below.

Before both experiments, participants completed an instruction and practice session to acquaint them with the task (~30 min). During practice, participants completed 100 shape-only trials and 16 prediction trials. The pair of auditory cues used during the short prediction run was not included in the main experiments.

After the experiments, participants completed a short questionnaire that indicated whether or not they became aware of the predictive nature of the auditory cues. The responses to both an open-ended question ("Can you tell us what the meaning of the sounds was during the experiment?") as well as a guided one ("During every block of the experiment, two different sounds were played. These sounds predicted which shapes would appear. For instance, a series of rising tones might predict that you'll see shape A, and falling tones might predict you'll see shape B. These predictions were 75% valid, so on 25% of trials they were incorrect. Did you realise this?") indicated that the vast majority of participants did not become aware of the predictions in either experiment (Experiment 1: 1 out of 22 participants indicated that they realised the cues predicted which shape would appear, no data for 2 participants; Experiment 2: 0 out of 22 participants indicated that they realised the cues predicted which shape would appear, no data for 2 participants).

**MRI acquisition**. In both experiments structural and functional MRI data were collected on a 3 T Siemens Prisma scanner with a 64-channel head coil at the Wellcome Centre for Human Neuroimaging (WCHN). Note that two different scanners with identical specifications were used for the two experiments, for availability reasons. Functional images for both experiments were acquired using a T2*-weighted multiband echo-planar imaging sequence (TR = 1000 ms; TE = 33.0 ms; 60 transverse slices; voxel size = 1.5 × 1.5 × 1.5 mm; flip angle = 55°,

multiband factor = 6). This sequence produced a partial volume for each participant, which covered the occipital and temporal lobes, including and parallel to the hippocampus. Field map data were acquired using a Siemens Field Map sequence (TR = 1020.0 ms; short TE = 10.00 ms; long TE = 12.46 ms; voxel size = 3.0 × 3.0 × 2.0 mm, 64 transverse slices, flip angle = 90°). Anatomical images were acquired using a T1-weighted Magnetisation Prepared Rapid Gradient Echo (MPRAGE), using a Generalized Auto calibrating Partially Parallel Acquisition (GRAPPA) factor of 2 (TR = 2530 ms; TE = 3.34 ms; 176 sagittal slices; voxel size = 1.0 × 1.0 × 1.0 mm; flip angle = 7°). To enable hippocampal segmentation, a T2-weighted turbo spin-echo (TSE) image (TR = 12650 ms; TE = 45 ms; voxel size = 0.4 × 0.4 × 1.5 mm; 54 coronal slices perpendicular to the long axis of the hippocampus; flip angle = 122°) was acquired.

**fMRI preprocessing**. Images for both experiments were preprocessed using Statistical Parametric Mapping (SPM12, http://www.fil.ion.ucl.ac.uk/spm, Wellcome Centre for Human Neuroimaging, London, UK). The first six volumes of each functional run were discarded to allow T1 equilibration. For each run, the remaining functional images were spatially realigned to correct for head motion, and simultaneously supplied to B0 unwarping, using SPM's realign and unwarp function. The functional data were temporally high-pass filtered with a 128 s period cut-off. No spatial smoothing was applied, and all analyses were performed in the participants' native space. The T1 and T2-weighted structural scans were co-registered and subsequently co-registered to the mean functional scan.

**Regions of interest**. The hippocampus and its subfields, CA1, CA2-3-DG, and the subiculum, were defined based on the structural T2 and T1 images using the automatic segmentation of hippocampal subfields (ASHS)[99] machine learning toolbox, in conjunction with a database of manual medial temporal lobe (MTL) segmentations from a separate set of 51 participants[100,101]. Consistent with previous studies, CA2, CA3 and DG were combined into a single region of interest (ROI) since these subfields are difficult to distinguish at our functional resolution (1.5 mm isotropic). This method also yielded an entorhinal cortex (EC) ROI for our informative connectivity analysis (see below). Results of the automated segmentation were inspected visually for each participant. In addition, a caudate region of interest (ROI), as well as visual cortex ROIs for our informational connectivity analysis—V1, V2 lateral occipital cortex (LO)—were automatically defined in each participant's T1-weighted anatomical scan using FreeSurfer (http://surfer.nmr.mgh.harvard.edu/). The visual cortex ROIs were restricted to the 500 most active voxels during the shape-only runs, to ensure that we were measuring responses in the retinotopic locations corresponding to our visual stimuli. Since no clear retinotopic organization is present in the other ROIs, cross-validated feature selection was used instead (see below). All ROIs were collapsed over the left and right hemispheres, as we had no hypotheses regarding hemispheric differences.

**fMRI data modelling**. For both experiments, the pattern of activity evoked by every single trial of the prediction runs, in each ROI, was estimated using the Least-Squares-Separate method[102,103]. That is, a separate GLM was created for every trial, such that each trial is modelled once as a regressor of interest, with all other trials combined into a single nuisance regressor. Delta functions were inserted at the onset of the trial of interest (first regressor) and all other trials (second regressor) and convolved with a double-gamma hemodynamic response function (HRF) and its temporal derivative[104]. The voxel-wise parameter estimates for the trial-of-interest HRF regressor constituted the estimated BOLD activity pattern for each trial. This method has been shown to improve the estimation of single-trial BOLD responses, compared with a GLM with one regressor for each trial[102]. In addition to these regressors, the GLMs included nuisance regressors consisting of the head motion parameters resulting from spatial realignment, their derivatives, and the square of these derivatives (i.e., 18 motion parameters in total). The data from the shape-only runs were analysed using a more conventional GLM, with one regressor for each of the five shapes and 18 head motion nuisance regressors.

**Shape decoding**. In order to probe neural shape representations, a forward modelling approach was used to decode the shapes from the patterns of BOLD activity in each ROI[27,105]. The decoding algorithm was identical to that used in Kok & Turk-Browne[27], and will be outlined here briefly (see Supplementary Fig. 1 for a visual depiction).

The shape selectivity of each voxel was characterised as a weighted sum of five hypothetical channels, each with an idealised shape tuning curve (or basis function), consisting of a halfwave-rectified sinusoid raised to the fifth power. In the first stage of the analysis, the parameter estimates obtained from the two shape-only runs were used to estimate the weights on the five hypothetical channels separately for each voxel, using linear regression. Specifically, let $k$ be the number of channels, $m$ the number of voxels, and $n$ the number of measurements (i.e., the five shapes). The matrix of estimated response amplitudes for the different shapes during the shape-only runs ($B_{train}$, $m \times n$) was related to the matrix of hypothetical channel outputs ($C_{train}$, $k \times n$) by a weight matrix ($W$, $m \times k$):

$$B_{train} = WC_{train} + N \tag{1}$$

The weight matrix was estimated by least squares estimation:

$$\widehat{\mathbf{W}} = \mathbf{B}_{train}\mathbf{C}_{train}^{T}(\mathbf{C}_{train}\mathbf{C}_{train}^{T})^{-1} \qquad (2)$$

Using these weights, the second stage of analysis consisted of reconstructing the channel outputs associated with the pattern of activity across voxels evoked by each trial in the prediction runs ($\mathbf{B}_{test}$), again using linear regression:

$$\widehat{\mathbf{C}}_{test} = (\widehat{\mathbf{W}}^{T}\widehat{\mathbf{W}})^{-1}\widehat{\mathbf{W}}^{T}\mathbf{B}_{test} \qquad (3)$$

where $\widehat{\mathbf{C}}_{test}$ are the estimated channel outputs. These channel outputs were used to compute a weighted average of the five basic functions, reflecting a neural shape tuning curve (Supplementary Fig. 1). Note that, during the main experiment (i.e., the prediction runs), only shapes 2 and 4 were presented. Decoding performance was quantified by subtracting the amplitude of the shape tuning curve at the presented shape (e.g., shape 2) from the amplitude at the non-presented shape (shape 4). This procedure yielded a measure of decoding evidence for the presented shape on each trial, in each ROI.

For all ROIs, voxel selection was based on data from the shape-only runs, in which no predictions were present, to ensure voxel selection was independent of the data in which we tested our effects of interest (i.e., the prediction runs). In visual cortex ROIs, we selected the 500 most active voxels during the shape-only runs. However, the hippocampus and caudate did not show a clear evoked response to visual stimuli, as defined by a lack of significant fit of a regressor of stimulus onset times convolved with a canonical haemodynamic response to the mean hippocampal time course. Therefore, we applied a different method of voxel selection for these ROIs. Voxels were first sorted by their informativeness, that is, how different the weights for the different channels were from each other, as indexed by the standard deviation of the weights. Second, the decoding model was trained and tested on different subsets of these voxels (between 10 and 100%, in 10% increments), within the shape-only runs (trained on one run and tested on the other). For all iterations, decoding performance on shapes 2 and 4 was quantified as described above, and the number of voxels that yielded the highest performance was selected. This procedure was used for voxel selection in the hippocampus (Experiment 1: 1068 voxels selected; Experiment 2: 970 voxels; group average), CA1 (Experiment 1: 271 voxels; Experiment 2: 313 voxels), CA2-3-DG (Experiment 1: 374 voxels; Experiment 2: 433 voxels), subiculum (Experiment 1: 273 voxels; Experiment 2: 249 voxels), and caudate (Experiment 1: 1292 voxels; Experiment 2: 1249 voxels).

**Quantifying time courses of shape representations**. A sliding window approach was used to investigate how shape representations evolved over trials. In Experiment 1, this window consisted of 4 trial positions (i.e., trials 1–4 of all 16 blocks, followed by trials 2–5, trials 3–6, etc.), while for Experiment 2 the window was four times as wide (16 trial positions; trials 1–16 of all 4 blocks, trials 2–17, trials 3–18, etc.) to compensate for the fourfold decrease in the number of blocks (i.e. the number of trials-per-position). Within each window, we averaged the decoding evidence for validly and invalidly predicted shapes separately. In order to quantify evidence for the shape predicted by the cue, controlling for the actually presented shape, evidence for validly and invalidly predicted shapes was subtracted (i.e., averaging (1 - evidence) for the invalidly predicted shapes with evidence for the validly predicted shapes) (Fig. 1e, f). Finally, the decoding time courses were smoothed by averaging over a sliding window. In Experiment 1 each bin was averaged with the previous and subsequent 4 bins, yielding a window size of 9 bins. In Experiment 2 the window size was 33 bins, containing the previous and subsequence 16 bins. Note that the results presented here do not critically depend on these parameters, as qualitatively identical effects were present when the length of the sliding window was doubled and subsequent smoothing was omitted. In the current study, analysing time courses without applying either a sliding window or temporal smoothing was not feasible, as fMRI responses to individual trials are not sufficiently robust. Future work could potentially address this by conducting multiple (e.g., four or more) fMRI sessions per participant, increasing the amount of data per trial position.

Initially, in Experiment 1, in a fully assumption-free analysis, we performed non-parametric cluster-based permutation tests[106] on the time courses, to test whether the decoding signals differed significantly from zero at any timepoint. Specifically, univariate $t$ statistics were calculated for all timepoints, and neighbouring elements that passed a threshold value corresponding to a $p$ value of 0.05 (two-tailed) were collected into clusters. Cluster-level test statistics consisted of the sum of $t$ values within each cluster, which were compared to a null distribution created by drawing 10,000 random permutations of the observed data. A cluster was considered significant when its $p$ value was below 0.05 (i.e., a cluster of its size occurred in fewer than 5% of the null distribution clusters). These non-parametric tests were specifically conceived to test effects in data with non-zero independence across time (and space), by generating null distributions with the same smoothness as the original data[106].

Subsequently, the obtained time courses of decoding evidence for the predicted shapes were quantified by fitting sigmoid curves to them. In Experiment 1, this consisted of a single sigmoid:

$$\frac{A}{1 + e^{-k(x-x_0)}} \qquad (4)$$

With a midpoint $x_0$ between trials 1 and 32, slope $k$ between 0.01 and 1, and amplitude $A$ between $-1$ and 1. These parameters were fitted using Matlab's fmincon function, wrapped in GlobalSearch. We ran 100 iterations with random parameter starting values (within their prescribed ranges), in order to avoid local minima. The amplitude parameter was submitted to simple $t$-tests to test whether learning curves significantly deviated from zero. In Experiment 2, the fitted curves consisted of a combination of two sigmoids, to test whether dynamics changed as learning progressed:

$$\frac{A_a}{1 + e^{-k_a(x-x_{0a})}} + \frac{A_b}{1 + e^{-k_b(x-x_{0b})}} \qquad (5)$$

with slopes $k_a$ and $k_b$ between 0.01 and 1, amplitudes $A_a$ and $A_b$ between $-1$ and 1. The first sigmoid had a midpoint $x_{0a}$ between trial 1 and 64, while the second had a midpoint $x_{0b}$ between trial 65 and 128, allowing them to capture potential differences between the first and second half of the blocks. Note that both sigmoids' amplitudes were free to range between $-1$ and 1, meaning that this analysis imposed no priors on the signs of the curves. As in Experiment 1, the amplitude parameters were submitted to simple $t$-tests to test whether learning curves significantly deviating from zero. Since Experiment 2 was motivated by a specific hypothesis on the nature of change of the hippocampal signal over trials (Fig. 4) we relied on these tests of the dynamics of the signal, rather than cluster-based permutation tests as in Experiment 1.

In an exploratory post hoc analysis, we also fitted two sigmoids to the data of Experiment 1. As in the analysis of Experiment 2, the first sigmoid was constrained to have a midpoint in the first half of the blocks (here, between trial 1 and 16), while the second had a midpoint in the second half (here, between trial 17 and 32), allowing them to capture potential differences between the first and second half of the blocks.

In a control analysis that made no assumptions on the shapes of the time courses, we calculated the average derivatives of the decoding time courses. For Experiment 2, this was done separately for the first (trials 1–64) and second (trials 65–128) half of the blocks, to investigate whether dynamics changed as learning progressed.

All analyses were initially performed on the hippocampus ROI as a whole, and when significant these were followed up by investigating hippocampal subfields and comparing the anterior and posterior hippocampi. This hierarchical approach, where significant effects in the ROI as a whole were followed up with tests of its subdivisions, rather than simply examining all possible comparisons, helped control the false positive rate. All statistical tests performed in this paper were two-sided.

**Informational connectivity**. In an exploratory analysis, we investigated whether functional connectivity between regions (specifically, between the posterior subiculum and EC, V1, V2, and LO) changed over trials in Experiment 2. Specifically, the Pearson correlation in decoding evidence over trials between two regions was calculated[107], within the sliding windows described above. This analysis yielded time courses of correlation values, with a positive value indicating that whenever region A represents shape 2 (rather than shape 4), region B is likely to do so as well. Changes in informational connectivity over time were tested by comparing $r$ values at the end of the blocks (i.e., the final window, containing trials 113–128) with the start of the blocks (the first window, containing trials 1–16), using paired-sample $t$-tests.

**Reporting summary**. Further information on research design is available in the Nature Research Reporting Summary linked to this article.

## Data availability
All region-specific fMRI time course data are available on the OSF platform (https://osf.io/48xjf/). Source data are provided with this paper.

## Code availability
All analysis scripts are available on the OSF platform (https://osf.io/48xjf/).

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

## Acknowledgements

The authors would like to thank Patricia Andrea Cabiles, Victoire Martignac and Ellis Langford for assistance with data collection, and Anna Schapiro for helpful discussion of these findings. This work was supported by a Wellcome/Royal Society Sir Henry Dale Fellowship [218535/Z/19/Z] and a European Research Council (ERC) Starting Grant [948548] to P.K. The Wellcome Centre for Human Neuroimaging is supported by core funding from the Wellcome Trust [203147/Z/16/Z].

## Author contributions

P.K. designed the study; F.A. collected the data; F.A. and P.K. analysed the data and wrote the manuscript.

## Competing interests

The authors declare no competing interests.
