## [Peer Review File · Nature Communications]

Hippocampal representations switch from errors to predictions during acquisition of predictive associationsREVIEWER COMMENTS

Reviewer #1 (Remarks to the Author):

Aitken and Kok report a pair of human fMRI studies that use a simple, implicit learning task to measure how stimulus representations in the hippocampus are influenced by predictability. In both experiments, auditory cues predicted an upcoming abstract shape with 75% validity. Subjects' task was simply to indicate whether the shapes (which were presented as a pair of successive images) changed size. From the subjects' perspective, the auditory cues were irrelevant. The only difference in experiment 1 vs 2 is that in experiment 1, cue-shape associations were presented for 32 trials and then reset whereas in experiment 2, they were presented for 128 trials, allowing for more time to learn the associations. Inverted encoding models were used to decode the identity of the shape from patterns of fMRI activity in the hippocampus. In experiment 1, towards the end of each learning block (i.e., after ~20-30 trials of learning) shape representations were relatively stronger for invalid than valid trials. In other words, shape information was stronger when the shape was unexpected compared to expected. In experiment 2, by considering a longer learning window, it was shown that this initial bias toward representing unexpected shapes went away and even reversed with more learning (such that there was a tendency toward stronger representation of expected over unexpected shapes). These results are framed in terms of the relative balance toward encoding versus retrieval and the relation of this balance to the information value (or 'newsworthiness') of current events.

Overall, this is an interesting paper of potentially broad interest to readers. The learning-related changes that are described are both novel and theoretically informative. The paper is very nicely written and the methods and results presentation are very clear. The inclusion of two fMRI studies is also a plus. However, I do have a few questions/concerns about whether some of the claims are fully supported by the data (or framed appropriately) and some suggestions for additional analyses. While the pattern of data is generally compelling, there are a few points where the data may not be quite as strong or as clear as the presentation suggests.

Comments:

1. In Experiment 1, the prediction error signal is observed at the end of the block (by trial 32). If anything, it looks like the effect maybe peaks by trial 28 or so. Yet, in experiment 2, there is absolutely no hint of this effect by trial 32. Instead, the peak is much later (~trial 60). Is there any explanation for this apparent discrepancy? Does this have something to do with the different sliding window/smoothing ranges across the studies? At a minimum, this discrepancy should be acknowledged.

2. It would be useful to see data from early visual cortex (V1, V2). Is the pattern of data similar to hippocampus? Why or why not? The only regions shown are hippocampal and caudate, and the general pattern is quite similar across these regions—which raises the question of whether there was any selectivity of these effects (i.e., other brain regions that did not contain information about prediction errors or predictions).

3. For Exp 2, the data qualitatively fit with the idea of a flip from prediction error to predictions. But, it is not clear that by the end of learning there is statistical evidence for valid>invalid in the hippocampus. In other words, for Figure 5B, is the line significantly above 0 at the end of learning? The positive sigmoid indicates that the early dip goes away. But, a positive sigmoid for the 2nd half could be obtained even if the curve simply returned to 0 (i.e., the dip went away). It does not appear that there is any direct test of whether the line(s) go above 0. I don't know that this issue is a potential deal breaker, but if there is no statistical evidence for valid>invalid, that is a fairly important caveat.

4. Relatedly, even if there is an effect of valid>invalid, it is not clear that this is a representation of the "predicted shape" as opposed to a representation of an expected shape. This is a somewhat subtle distinction—and the authors use both terms—but calling it a representation of a predicted shape implies that it is a pure representation of prediction. But, with the current design, on the valid trial the predicted shape obviously matches the actual shape and there is no way to temporally tease apart the prediction that may have been generated from the encoding of the current stimulus. Thus, it is as much prediction as it is perception. As such, it feels more appropriate to describe these data in terms of whether a stimulus was expected or predictable as opposed to a representation of the "predicted shape."

5. That said, for invalid trials, it does seem that the authors could test for an actual prediction effect. That is, if stimulus A is invalidly cued (and the outcome is actually B), then it seems that evidence for stimulus A could be compared against evidence for the other 3 shapes (from the shape only runs). For the invalid trials, is the evidence for A greater than for C, D, E? Does this have any relation to the prediction error effect that is reported?

6. There are a few points related to the temporal smoothing that I think could use some justification or explanation:

- Are the cluster level statistics appropriate given the strong non-independence across time points that is induced by the sliding window and temporal smoothing?

- Am I correct that decoding was applied using data from multiple trials (4 in Exp 1, 16 in Exp 2) in a sliding window format and then AFTER this, temporal smoothing was also performed? Why both steps (sliding window + smoothing)?

- It would be good to make clear in the main text/figures that a sliding window was used for the analyses. The data are unnaturally smooth over time and this really jumped out to me when I first saw the data.

Reviewer #2 (Remarks to the Author):

Aitken & Kok report an interesting study about hippocampal contributions to statistical learning over

time. In this fMRI study, participants learned (implicitly) that auditory cues probabilistically predicted specific shapes. After some initial experience with these statistical regularities, hippocampal patterns began to represent prediction errors (i.e., when the outcome shape was surprising, this outcome would be represented more strongly). However, after extensive experience, hippocampal patterns instead represented the predicted shape regardless of the outcome.

The authors interpret this key finding as a shift from signalling errors, which are most informative during initial learning, to signalling established predictions, which is valuable when learning is “complete.” The idea that errors should be discounted when they can be attributed to random noise or expected uncertainty is intriguing. The connectivity analyses were also theoretically interesting (suggesting a shift in connectivity with visual input regions).

Overall, the results are novel and inform to key ongoing debates in the cognitive neuroscience of memory. This has the potential to be an important contribution to the literature. However, in its present form, it was difficult to evaluate some of the claims because elements of the analysis and approach were not clear. These aspects need to be clarified if the paper is to make a broad impact. These comments are detailed below.

Major Points:

1. The timing of the trial-wise multivariate analysis is unclear.

Figure 1F implies that the classifier should detect very different representations during the prediction phase vs. the prediction error phase, which makes sense. However, the methods (in the main text and extended methods) do not clearly state when (within each trial) activation patterns were analyzed.

It seems as though the activation patterns for each trial may have been averaged across the entire trial duration, including both the prediction phase and the outcome phase. These distinct phases were also quite brief (500 ms delay between cue and outcome), which may have made it difficult to model prediction and outcome phases separately.

If activation patterns were indeed averaged across the entire trial, the motivation behind this design choice is not clear. The brain should generate a prediction after the cue first arrives, then process the outcome (whether expected or surprising) when it is presented later. As shown in Figure 1F, the different trial types should lead to different representations as each trial progresses. I understand that Valid and Invalid trials were compared so that the outcome shape was the same (i.e., only the prediction phase differed). However, it seems that on Invalid trials, the evidence for B (during the prediction phase) would be negated by the evidence for B (during the subsequent outcome phase). See also major point #2.

Overall, it seems difficult to draw clear conclusions about whether the hippocampus is primarily responsible for representing prediction vs. prediction error if this analysis collapses across both prediction and outcome phases of each trial. It is likely that both prediction and prediction error signals are present on every trial, but the relative strength of the two signals evolves over time. This

idea is briefly mentioned in the Discussion, but the main interpretation throughout the paper implies a binary flip from a “learning mode” to a “predicting mode.” I am not sure that this mode-switching idea is sufficiently supported by the evidence.

If activation patterns were not averaged across the whole trial, it would be helpful to explain this approach in more detail to prevent confusion.

2. The strategy of subtracting classifier evidence for Valid – Invalid trials is unclear.

Related to the first major point above, I found other aspects of Figure 1 difficult to interpret. In particular, the strategy of subtracting classifier evidence for Valid – Invalid trials is not clear; this is important for understanding the results and could be explained more clearly in the methods and figure. I was confused by the correspondence between panels 1E and 1F. It seems that the reader is supposed to look across each color-coded row.

The top row in green indicates that when A is expected and A occurs, we should be able to decode the prediction for A, but not B (because no prediction error occurred). This row makes sense. The middle row in red indicates that when B is expected but A occurs, we should be able to initially decode the prediction for B, but this should switch to A when the unexpected outcome appears. This row is confusing if activation patterns were actually averaged across the whole trial; wouldn't the evidence for A and B cancel out?

The bottom row in yellow is also confusing. On the right, the “predicted shape effect” is defined as the difference between A – B trials during the prediction phase (controlling for the same outcome). On the right, we see the net result for decoder evidence after this subtraction step. I was under the impression that the 1F should now be read from top to bottom: A (Valid, green bar) is subtracted from B (Invalid, red bar) to yield the yellow bar. The “Prediction error” column in 1F aligns with this logic. However, the “Prediction” column does not. Why is the net result (yellow bar) strong evidence in favor of A?

3. There are discrepancies between the results for Exp 1 and Exp 2 that are not discussed sufficiently.

It seems that the only difference between Exp 1 and Exp 2 is the number of trials per block (32 trials in Exp 1 vs. 128 trials in Exp 2). The authors state that the allocation of trials to blocks is the only difference between the two experiments. Therefore, we should see the same results from Exp 1 recapitulated in the first ~32 trials of Exp 2. Figure 1 demonstrates that in Exp 1, the effect of Invalid > Valid classifier evidence (the “prediction error representation”) emerged after trial 22.

However, in the Exp 2 results, it looks like the effect of Invalid > Valid only emerged after trial 40 (Figure 5). That is more trials than the entire block in Exp 1. This suggests that learning (the “prediction error” representation stage) was slower and took longer in Exp 2. This discrepancy between the two experiments should be discussed.

One possibility is that this discrepancy arises from differences in the sliding window analysis (averaging classifier timecourses across bins of trials). In Exp 1, the bins included 4 trials. In Exp 2, the bins were expanded to 16 trials. The authors state that the bin size was increased because each block was 4x longer in Exp 2 relative to Exp 1. However, it's not clear why it was necessary to equate the number of bins across experiments. Wouldn't it be more straightforward to compare results across the two experiments if the bin size was the same, but Exp 2 showed more bins (on a longer x-axis)? This would more clearly show whether the Exp 1 timecourse was replicated in Exp 2, before the Exp 2 results diverged as learning progressed and plateaued.

4. There appear to be effects that do not necessarily align with the hypotheses and are not discussed.

The Exp 1 statistics reported pertain to the Invalid > Valid difference that emerges in the trials at the end of each block. However, Figure 2 suggests that there is also an effect in the opposite direction (Valid > Invalid) in the middle of each block. This effect is most evident for the Subiculum, but also appears to a lesser extent in the CA1 and whole-hippocampus plots.

This opposite effect should be acknowledged in the Results, and statistics should be reported for the earlier portions of the learning blocks. Is there an explanation for why this Valid > Invalid effect might occur during the middle phase of learning? If substantial, unexplained fluctuations in decoding evidence can occur in either direction, it casts doubt on the reliability of the measure (especially because the conclusions depend on interpreting fluctuations over trials).

Minor Points:

1. The Introduction does not clearly motivate the main conclusion of the paper.

The introduction states that the hippocampus signals prediction errors (i.e., mismatch signals), but can also represent predictions. This statement is framed as a contradiction that must be resolved, but it is not clear why these two functions would be contradictory. Prediction and error can occur at different timepoints (even within the same trial) without being in conflict. Assuming there is some relevant prior knowledge to guide a prediction, the hippocampus should represent that prediction until the feedback/outcome/stimulus arrives. At that point, it should identify whether the outcome matches or mismatches the expectation.

It would be helpful if the introduction more clearly motivated the key idea that once learning is "complete" the hippocampus should be somewhat resilient to encountering random noise. After extensive prior experience, perhaps prediction errors should not lead us to radically revise our models of the world. If some uncertainty/noise in outcomes is to be expected, prediction error ceases to be a learning signal. This message is clear in the Discussion, but is not laid out clearly in the Introduction.

Reviewer #3 (Remarks to the Author):

This paper describes two fMRI experiments applying MVPA in the human hippocampus (and subfields) during an auditory-visual contingency learning procedure. Experiment 1 shows that hippocampal regions initially start to code unexpected visual stimuli (which occur on 25% of trials). Experiment 2 uses more trials (128 rather than 32) and replicates the initial hippocampal coding of unexpected stimuli, but then shows that hippocampus later switches to code expected stimuli, regardless of the actual stimuli. The conclusion is that hippocampus initially responds to prediction errors, but once their occurrence is learned, they are “no longer treated as model updating (‘newsworthy’) events, are therefore no longer upweighted and the retrieval of the cued shape dominates”, consistent with some previous studies.

The analyses seem well-conducted, though I have a few comments, and I note that the claims would be stronger still if they replicated the “double-exponential” shape in Experiment 2 at least once.

1. The authors claim that the behavioural data (accuracy/RTs) for the shape-matching task are unaffected by the validity of the auditory cue (since this is not directly task-relevant). However, it seems that a surprising first shape could disrupt performance, at least until the proportion of such surprises is learned (as part of the “environment model”). So rather than averaging over trials, could the authors analyse behavioural data as a function of trial, like they did for the fMRI data?

2. Where are the horizontal bars indicating cluster-level correction for Experiment 2 (ie Figure 5), or did differences never survive correction? I appreciate that the authors might prefer inferences based on fitting their sigmoid functions (which is probably more sensitive), but this doesn't allow such precise estimation of when significant decoding happens. This is important, because in Experiment 1, significant decoding of the unexpected shape occurs after about 24 trials, whereas in Experiment 2, it doesn't look like this occurs until 50 trials, i.e. twice as long. Why is this? I could not see an obvious procedural difference between experiments that would explain this. (Better temporal localisation would be achieved by correcting for peak statistic rather than cluster, but this is even less likely to reveal significance.)

3. The switch to coding the expected stimulus after about 100 trials in Experiment 2 is certainly very interesting, and reconciles previous findings from the same group (Kok & Turk-Browne), but it is still a novel finding (I think) within a continuous analysis learning procedure, so it would be more convincing if it were replicated at least once, ideally with all analyses pre-registered, eg on OSF.

4. Would it be helpful to have an extra control condition, to control for “time-within-block” (e.g. which might affect interest/attention to the auditory cues). Such a condition could have no clear auditory-visual contingency for example, so would not be expected to show much decoding.

Minor points

5. In Experiment 1, there is an earlier divergence between expected and unexpected stimuli around 16 trials (at least in hippocampus and subiculum), favouring decoding of expecting stimuli. I appreciate that this does not reach significance, and is not obvious in Experiment 2, and therefore likely to be noise, but would it be significant if fit with a second, even earlier sigmoid in Experiment 1? It would be

reassuring to know that the sigmoidal fitting is robust to overfitting such potential noise.

6. Is correction for multiple comparisons needed across the number of ROIs (subfields)?

7. Extraneous “bar” after ref 30 on line 59

Thank you for providing the data and code. I did not have time to run it all, but it did help me confirm aspects of the analysis like the randomisation used for cluster-level correction.

We would like to thank the reviewers for their thorough and constructive assessment of our manuscript. We have revised the manuscript to address their points of critique, which we believe has led to a greatly improved paper. Our point-by-point responses are detailed below.

Reviewer #1

1. In Experiment 1, the prediction error signal is observed at the end of the block (by trial 32). If anything, it looks like the effect maybe peaks by trial 28 or so. Yet, in experiment 2, there is absolutely no hint of this effect by trial 32. Instead, the peak is much later (~trial 60). Is there any explanation for this apparent discrepancy? Does this have something to do with the different sliding window/smoothing ranges across the studies? At a minimum, this discrepancy should be acknowledged.

We thank the reviewer for raising this point; apologies for not discussing this discrepancy in the previous version of the manuscript. We agree with the reviewer that it is likely the result of the differences in sliding window length and smoothing between the two experiments. That is, the sliding window and smoothing kernel were four times longer in Experiment 2 than in Experiment 1. This was necessary to achieve a similar signal-to-noise ratio in both experiments; there were four times fewer blocks in Experiment 2, since the blocks were four times longer and the fMRI session could not be made any longer than it already was.

Using a sliding window of four trials would have resulted in only $4 \times 4 = 16$ trials per bin in Experiment 2, of which only 4 (25%) would have been invalid, two per shape. Needless to say, this would have resulted in very noisy fMRI signal estimates, especially for the crucial invalid cue trials. (See also our reply to point 6 below for more details on the temporal smoothing implemented here.)

Note that Fig. 2b suggests that the negative predicted shape evidence is still decreasing at the end of the 32 trial blocks, as a result of the valid shape evidence (Fig. 2a) becoming increasingly negative, suggesting that perhaps the peak of the negative signal indeed lies later than at the end of the blocks in Experiment 1.

Speculatively, the delayed negative peak in Experiment 2 may also partly have resulted from averaging together a small positive peak around trial 16 (Fig. 2a, b; see especially the subiculum in Fig. 2d and a new analysis presented in Supplementary Fig. 3) with the subsequent negative signal. Whereas the sliding window in Experiment 1 may have been short enough to separately resolve these two signals, the longer sliding window in Experiment 2 was not, resulting in these two signals cancelling each other out early in the blocks. Note that this initial positive signal in Experiment 1 was quite weak, and only detectable using post-hoc analyses (see response to Reviewer 2 point 4 for details), making this explanation very speculative. We nevertheless now include it in the Discussion (pp. 14-15), to acknowledge the discrepancy between the two experiments and suggesting a way future research may be able to shed more light on this:

“An exploratory post-hoc analysis of Experiment 1 additionally revealed early prediction-like signals in the subiculum, before the prediction error-dominated signals emerged (Supplementary Fig. 3). This initial positive signal could potentially reflect early, imprecise predictions, which lead to strong prediction errors on subsequent invalid trials. This explanation is currently speculative, especially given the post-hoc nature of the analysis. Future research is needed to investigate the early build-up of prediction signals further, for instance using a paradigm with many blocks with only a few cue repetitions each.

It appears that the prediction error-like signals emerged later in Experiment 2 than in Experiment 1 (cf. Fig. 2 and 5). This is likely the result of differences in sliding window length and smoothing between the two experiments, combined with the fact that the prediction error signal may still be building by the end of the blocks in Experiment 1 (Fig. 2b). More speculatively, the later negative peak in Experiment 2 may also partly have resulted from averaging together the small early positive peak discussed above (~trial 16-20) with the subsequent negative signal (~trial 24 onwards) (Supplementary Fig. 3). While the sliding window in Experiment 1 was short enough to separately resolve these two signals, the longer sliding window in Experiment 2 was not, resulting in these two signals cancelling each other out early in the blocks. At present this explanation is highly speculative since the initial positive signal in Experiment 1 was detected using post-hoc analyses and needs to be investigated further, as discussed above.”

2. It would be useful to see data from early visual cortex (V1, V2). Is the pattern of data similar to hippocampus? Why or why not? The only regions shown are hippocampal and caudate, and the general pattern is quite similar across these regions—which raises the question of whether there was any selectivity of these effects (i.e., other brain regions that did not contain information about prediction errors or predictions).

Thank you for the suggestion of investigating these additional brain regions. We have now conducted these analyses, and they did not reveal similar prediction-related dynamics as the hippocampus and caudate (now included in Supplementary Fig. 8b-d, reproduced below for convenience). It is noteworthy that there is a numerical trend towards the emergence of a prediction-like signal towards the end of the long blocks in early visual cortex (V1 and V2), in line with the results in Kok & Turk-Browne (2018), but these trajectories did not reach significance when probed with a sigmoid fit (all $p > 0.05$). (Though note that there was significantly positive prediction effect in the final bin in V2 ($t_{23} = 2.18, p = 0.04$) and a trend in V1 ($t_{23} = 1.89, p = 0.072$). This seems in line with the fact that early visual cortex revealed fairly subtle (but significant) prediction effects in a similar paradigm where predictions were learnt before scanning commenced (Kok & Turk-Browne 2018), but these regions were more strongly dominated by presented rather than predicted visual stimuli.

Further, we investigated representations in the amygdala, a region adjacent to the hippocampus that we did not expect to play a role in these perceptual predictions. Indeed, the amygdala did not reveal prediction error or prediction-like signals over the course of learning (Supplementary Fig. 8e), despite its close proximity to the hippocampus. In sum, the dynamic prediction error and prediction-like signals revealed in the current study appear to be quite selective to the hippocampus and the caudate nucleus, in line with a searchlight analysis conducted in Kok & Turk-Browne 2018 (their Fig. 3C). We believe that adding these results to our manuscript has strengthened the paper and we thank the reviewer for this suggestion.

Supplementary Fig. 8. Experiment 2 shape decoding over trials in additional regions. Decoding evidence for validly (green) and invalidly (red) predicted shapes (left panel), decoding evidence for predicted (valid – invalid) shapes (middle panel), and amplitude parameters of early (midpoint between trials 1 and 64) and late (midpoint between

trials 65 and 128) sigmoid curves (right panel) in **a** the caudate nucleus, **b** V1, **c** V2, **d** Lateral occipital cortex, **e** Amygdala. Shaded regions and error bars indicate SEM. Dots indicate individual participants. ** $p < 0.01$.

3. For Exp 2, the data qualitatively fit with the idea of a flip from prediction error to predictions. But, it is not clear that by the end of learning there is statistical evidence for valid>invalid in the hippocampus. In other words, for Fig. 5B, is the line significantly above 0 at the end of learning? The positive sigmoid indicates that the early dip goes away. But, a positive sigmoid for the 2nd half could be obtained even if the curve simply returned to 0 (i.e., the dip went away). It does not appear that there is any direct test of whether the line(s) go above 0. I don't know that this issue is a potential deal breaker, but if there is no statistical evidence for valid>invalid, that is a fairly important caveat.

We agree with the reviewer that it is important to test whether the signal flips from negative to positive rather than going from negative to zero. We previously included statistical tests of the signal versus zero at the end of the blocks in Experiment 2 in subdivisions of the hippocampus (posterior vs. anterior, subfields), but we have now substantially expanded this (pp. 12-13):

“A positive slope in the second half of the blocks might also be observed if the early prediction error signal simply gradually disappeared, rather than hippocampal representations switching to a positive prediction signal. To resolve this, we tested whether decoding evidence for the predicted shape at the end of the blocks (i.e., the final bin) was significantly larger than zero. While this signal was not significantly positive for the hippocampus as a whole ($t_{23} = 1.75$, $p = 0.094$), it was in the posterior hippocampus ($t_{23} = 2.15$, $p = 0.042$), which was also the driver of the results reported above. In fact, decoding evidence for the predicted shape at the end of the blocks was stronger in the posterior than the anterior ($t_{23} = 0.31$, $p = 0.76$; posterior vs. anterior: $t_{23} = 2.14$, $p = 0.043$) hippocampus. The significant effect in posterior hippocampus was reflected by significant evidence for the predicted shape in the posterior subiculum ($t_{23} = 2.55$, $p = 0.018$) and CA1 ($t_{23} = 2.17$, $p = 0.041$), but not CA2-3-DG ($t_{23} = 1.31$, $p = 0.20$). [...] Note that this analysis was based on a relatively small subset of trials (i.e., only those in the last bin of each block), but the positive evidence for predicted shapes and absence of evidence for the presented shapes is in line with a study employing a virtually identical paradigm where participants learnt the cue-shape predictions before being tested in the scanner²⁷. There was no significant difference in predicted shape evidence between the posterior subfields ($F_{2,46} = 1.09$, $p = 0.34$), but it is worth noting that the effect was numerically largest in the posterior subiculum (0.13; Supplementary Fig. 7), in line with previous work²⁷.

[...]

The caudate nucleus displayed [...] a significantly positive representation of the predicted shape in the final bin of the blocks ($t_{23} = 3.06$, $p = 0.0055$).”

We are encouraged by the fact that even though this analysis is based on a relatively small subset of trials, the results are in line with a recent study where participants learnt the cue-shape predictions before being tested in the scanner, i.e., participants were essentially in a similar state as at the end of the blocks here for the full experiment (Kok & Turk-Browne 2018).

4. Relatedly, even if there is an effect of valid>invalid, it is not clear that this is a representation of the “predicted shape” as opposed to a representation of an expected shape. This is a somewhat subtle distinction—and the authors use both terms—but calling it a representation of a predicted shape implies that it is a pure representation of prediction. But, with the current design, on the valid trial the predicted

shape obviously matches the actual shape and there is no way to temporally tease apart the prediction that may have been generated from the encoding of the current stimulus. Thus, it is as much prediction as it is perception. As such, it feels more appropriate to describe these data in terms of whether a stimulus was expected or predictable as opposed to a representation of the “predicted shape.”

We acknowledge that we cannot temporally tease apart pre-stimulus from post-stimulus signals at the temporal resolution afforded by fMRI, and we now acknowledge this explicitly in the manuscript (p. 6 and p. 14; text copied below for convenience). However, we do feel our results go beyond a simple effect of valid > invalid, namely in the sense that on invalid trials there is actually evidence for the cued/predicted shape rather than the presented one. This is related to point 5 below, where we discuss this in more detail and present new results on the representation of invalidly predicted shapes. In other words, it is not the case that both validly and invalidly predicted shapes are represented but valid shapes are simply represented more strongly (cf. effects of prediction in visual cortex; Kok et al. Neuron 2012; Yon et al. Nat Comm 2018), but instead there is no significant representation of the presented shape (which is identical on valid and invalid trials, and can thus be quantified by $(\text{valid} + \text{invalid})/2$) at all, but only of the shape predicted by the cue ($\text{valid} - \text{invalid}$). We realise now that we did not explain this subtraction logic sufficiently well in the previous version of the manuscript, and we did not in fact include statistic tests of the effect of ‘presented shape’ (as we did in Kok & Turk-Browne 2018, see e.g. their Fig. 2a-c). We have now included this (p. 6):

“Multivoxel decoding analyses (Supplementary Fig. 1), trained on data from separate shape-only runs in which no predictive cues were presented (Fig. 1c, d), were used to reveal hippocampal shape representations on valid and invalid trials (Fig. 1e). If the hippocampus were to represent prediction errors, valid trials should not result in a shape representation, since the predicted and presented shapes are identical and should cancel each other out (Fig. 1f, top left). On invalid trials on the other hand, if shape B is predicted but shape A is presented, unexpected shape A should be represented in the hippocampus (Fig. 1f, middle left). If instead the hippocampus were to represent predictions rather than errors, on invalid trials where shape B is predicted but shape A is presented, shape B should be represented in the hippocampus (Fig. 1f, middle right). Further, on valid trials the shape that is both predicted and presented should be represented (Fig. 1f, top right).

Both of these types of patterns have been observed in the hippocampus⁵⁶, and the aim of the current study was to investigate how they develop over the course of learning. Note that the temporal resolution afforded by fMRI did not allow us to investigate any potential fast within-trial dynamics of these hypothesised prediction and prediction error signals. Rather, the shape representations revealed here reflect a temporal integration of neural signals over the course of a trial. It seems likely that both predictions and prediction errors play a role in hippocampal computations, the question addressed here is whether the relative weighting of the two is affected by novelty and uncertainty.

The clearest way to dissociate effects of the predictive cues from the effects of the presented shapes is to subtract decoding evidence for the invalidly predicted shapes from evidence for the validly predicted shapes (Fig. 1e), since the presented shapes were identical in both types of trials. Under a prediction error hypothesis, this would result in a negative signal (subtracting a positive signal on invalid trials from a zero signal on valid trials; Fig. 1f left column). Under a prediction hypothesis on the other hand this would result in a positive signal (subtracting a negative signal on invalid trials from a positive one on valid trials; Fig. 1f right column). This subtraction therefore constitutes our main effect of interest. Additionally, averaging the evidence for validly and invalidly predicted shapes allowed us to quantify evidence for the shape as presented on the screen, regardless of the cues²⁷.”

And on p. 12:

“There was no significant evidence for the presented shape (quantified by averaging evidence for valid and invalid shapes, thereby averaging out the effect of the cues²⁷; Fig. 1e) at the end of the blocks in the hippocampus ($t_{23} = -1.30$, $p = 0.21$) or any of its subdivisions (all $p > 0.2$). In other words, once predictions were learnt, hippocampal representations were determined by the predictive cues, not by which shape was actually presented on screen.”

And in the Discussion, on p. 14:

“Similarly, the slow nature of the BOLD signal prevents investigating fast within-trial dynamics of prediction and prediction error signals. For instance, it may be that prediction signals always precede prediction error signals in the hippocampus. The hippocampal representations revealed here reflect a temporal integration of neural signals over the course of a trial, and thus indicate whether predictions or prediction errors dominate. Future studies with millisecond temporal resolution are required^{9,82} to reveal the dynamic interplay of predictions and errors within the hippocampus.”

As the reviewer has gleaned, these effects most crucially rely on the shape representations during the invalidly predicted trials, which we go into in more detail in reply to point 5 below.

5. That said, for invalid trials, it does seem that the authors could test for an actual prediction effect. That is, if stimulus A is invalidly cued (and the outcome is actually B), then it seems that evidence for stimulus A could be compared against evidence for the other 3 shapes (from the shape only runs). For the invalid trials, is the evidence for A greater than for C, D, E? Does this have any relation to the prediction error effect that is reported?

We thank the reviewer for this excellent suggestion. We have now unpacked the shape representations on the invalid trials by not only considering the subtraction of evidence between the presented and non-presented shape, but visualising evidence for all five possible shapes. In line with a prediction effect, we find that at the end of the blocks evidence on invalid trials is stronger for the (invalidly) cued shape than for the other shapes, including the presented shape. Halfway through the block on the other hand, evidence was strongest for the shape that was (unexpectedly) presented on the screen. These new results are described in the Results (p. 10):

“This switch was most striking for invalidly predicted shapes (Supplementary Fig. 5a). Initially, approximately halfway through the blocks, the unexpectedly presented shape was represented (Supplementary Fig. 5b). However, at the end of the blocks, the hippocampus instead represented the shape predicted by the auditory cue, rather than the shape presented on the screen (Supplementary Fig. 5b).”

And displayed in the new Supplementary Fig. 5, reproduced below for convenience. We believe these new results enrich the paper and thank the reviewer for suggesting this analysis.

Supplementary Fig. 5. Evolution of shape decoder output on invalid trials in the hippocampus. **A** Shape decoder output for invalidly predicted shapes, over trials (x-Axis), across the full range of possible shapes as presented in the shape-only runs (y-Axis). See Supplementary Fig. 1 and Methods for more details on the shape decoding algorithm. Note that these are the same data underlying the red time course in Fig. 5a, which consists of the subtraction of evidence for the presented shape (here, shape 2) minus the non-presented shape (here, shape 4, in this case the predicted shape). **B** Two vertical slices from panel A; decoder output for invalid shapes halfway through the blocks (trials 56-72, top panel) and at the end of the blocks (trials 116-128, bottom panel). Halfway through, evidence for the presented shape (shape 2) was stronger than for the shapes 1 ($t_{23} = 2.22$, $p = 0.036$), 4 (the predicted shape; $t_{23} = 2.59$, $p = 0.016$), and 5 ($t_{23} = 3.17$, $p = 0.0043$), but not for shape 3 ($t_{23} = 1.06$, $p = 0.30$). At the end of the blocks, evidence for the predicted shape (shape 4) was stronger than for shapes 1 ($t_{23} = 2.60$, $p = 0.016$) and 2 (the presented shape; $t_{23} = 2.07$, $p = 0.050$), but not significantly different from evidence for shapes 3 ($t_{23} = 1.78$, $p = 0.088$) and 5 ($t_{23} = 0.92$, $p = 0.37$). Shaded regions indicate SEM.

6. There are a few points related to the temporal smoothing that I think could use some justification or explanation:

- Are the cluster level statistics appropriate given the strong non-independence across time points that is induced by the sliding window and temporal smoothing?

Yes, the non-parametric cluster-based permutation tests we used were in fact specifically conceived to handle data with non-zero independence across time (and space) (Maris & Oostenveld, Journal of Neuroscience Methods, 2007). That is, the permuted data have the same smoothness as the non-permuted data, and therefore the null distribution contains temporal clusters that would be expected by chance from data with this level of smoothness. Only clusters that are larger than would be expected by chance ($p < 0.05$) are considered statistically significant. We have now amended the Methods to elucidate this important point (p. 21):

“Initially, in Experiment 1, in a fully assumption-free analysis, we performed non-parametric cluster-based permutation tests¹⁰⁶ on the time courses, to test whether the decoding signals differed significantly from zero at any timepoint. Specifically, univariate t statistics were calculated for all timepoints, and neighbouring elements that passed a threshold value corresponding to a p value of 0.05 (two-tailed) were collected into clusters. Cluster-level test statistics consisted of the sum of t values within each cluster, which were compared to a null distribution created by drawing 10,000 random permutations of the observed data. A cluster was considered significant when its p value was below 0.05 (i.e., a cluster of its size occurred in fewer than 5% of the null distribution clusters). These non-parametric tests were

specifically conceived to test effects in data with non-zero independence across time (and space), by generating null distributions with the same smoothness as the original data¹⁰⁶.”

- Am I correct that decoding was applied using data from multiple trials (4 in Exp 1, 16 in Exp 2) in a sliding window format and then AFTER this, temporal smoothing was also performed? Why both steps (sliding window + smoothing)?

That is correct. A degree of smoothing was necessary to have sufficient signal-to-noise ratio for our analyses, since whereas most fMRI experiments average trials over a full session, here we are investigating dynamics over time. We chose to do this in the way the reviewer describes, but we acknowledge that smoothing could have been implemented in other ways as well. In a new control analysis, we doubled the length of the sliding windows (8 trials in Exp 1, 32 trials in Exp 2) and omitted the temporal smoothing afterwards. As can be seen in Fig. R1, this yielded qualitatively identical effects as our main analysis (cf. Fig 2a and Fig. 5a).

This confirms that our results are robust against the exact procedure by which the data are smoothed, and we now mention this in the Methods (p. 21):

“Note that the results presented here do not critically depend on these parameters, as qualitatively identical effects were present when the length of the sliding window was doubled and subsequent smoothing was omitted.”

We suggest not to include this figure in the manuscript, as we already have quite a few (main and supplemental) figures – however, if the reviewer feels we should, we are happy to oblige.

Fig. R1. Shape decoding over trials in both experiments, longer sliding window but no smoothing afterwards. Left panels: Decoding evidence for validly (green) and invalidly (red) predicted shapes in the hippocampus in Experiment 1 (top) and Experiment 2 (bottom). Middle panels: Decoding evidence for predicted (valid – invalid) shapes in hippocampus in hippocampus in Experiment 1 (top) and Experiment 2 (bottom). In Experiment 1, cluster-based permutation tests reveal significant evidence for invalidly predicted shapes ($p = 0.041$) and significantly

negative evidence for predicted shapes ($p = 0.036$) towards the end of the blocks (cf. Fig. 2a). This was reflected by a significantly negative sigmoid fitted to the predicted shape evidence over trials ($t_{23} = -2.19$, $p = 0.039$). In Experiment 2, an initial negative sigmoid ($t_{23} = -2.11$, $p = 0.046$) was followed by a positive sigmoid ($t_{23} = 2.51$, $p = 0.019$) (cf. Fig. 5a).

- It would be good to make clear in the main text/figures that a sliding window was used for the analyses. The data are unnaturally smooth over time and this really jumped out to me when I first saw the data.

We apologise for not making this sufficiently clear in the previous version of the manuscript. We have now included statements to this effect in the main text and the figure legends. E.g., on p. 7, at the start of the 'fMRI decoding results' section:

"The dynamics of hippocampal shape representations over trials were investigated using a sliding window approach (see Methods for details)."

And in the figure legends (e.g. in the legend of Figure 2):

"Time courses were temporally smoothed using a sliding window approach (see Methods for details)."

Reviewer #2

Major points

1. *The timing of the trial-wise multivariate analysis is unclear.*

Fig. 1F implies that the classifier should detect very different representations during the prediction phase vs. the prediction error phase, which makes sense. However, the methods (in the main text and extended methods) do not clearly state when (within each trial) activation patterns were analyzed.

It seems as though the activation patterns for each trial may have been averaged across the entire trial duration, including both the prediction phase and the outcome phase. These distinct phases were also quite brief (500 ms delay between cue and outcome), which may have made it difficult to model prediction and outcome phases separately.

If activation patterns were indeed averaged across the entire trial, the motivation behind this design choice is not clear. The brain should generate a prediction after the cue first arrives, then process the outcome (whether expected or surprising) when it is presented later. As shown in Fig. 1F, the different trial types should lead to different representations as each trial progresses. I understand that Valid and Invalid trials were compared so that the outcome shape was the same (i.e., only the prediction phase differed). However, it seems that on Invalid trials, the evidence for B (during the prediction phase) would be negated by the evidence for B (during the subsequent outcome phase). See also major point #2.

Overall, it seems difficult to draw clear conclusions about whether the hippocampus is primarily responsible for representing prediction vs. prediction error if this analysis collapses across both prediction and outcome phases of each trial. It is likely that both prediction and prediction error signals are present on every trial, but the relative strength of the two signals evolves over time. This idea is briefly mentioned in the Discussion, but the main interpretation throughout the paper implies a binary flip from a “learning mode” to a “predicting mode.” I am not sure that this mode-switching idea is sufficiently supported by the evidence.

If activation patterns were not averaged across the whole trial, it would be helpful to explain this approach in more detail to prevent confusion.

We apologise for not making the design logic sufficiently clear. As the reviewer gleaned, activation patterns studied here using fMRI reflected an average cross the whole trial. We now make this explicit in the manuscript (p. 6):

“Note that the temporal resolution afforded by fMRI did not allow us to investigate any potential fast within-trial dynamics of these hypothesised prediction and prediction error signals. Rather, the shape representations revealed here reflect a temporal integration of neural signals over the course of a trial. It seems likely that both predictions and prediction errors play a role in hippocampal computations, the question addressed here is whether the relative weighting of the two is affected by novelty and uncertainty.”

We chose to use fMRI since it afforded us the high spatial resolution required to resolve activity patterns in the hippocampus and its subfields. The limited temporal resolution of fMRI means that the only way to distinguish the cue-phase from the outcome-phase would be to either make the interval between cue and stimulus very long (10+ seconds) or jitter it substantially (e.g. 4-10s), both of which would have likely resulted in a much weaker predictive association between cue and outcome. And even then, separation of BOLD responses to cues and outcomes would have been imperfect. Therefore, we chose to refrain from temporally dissociating cue and outcome phases using fMRI, relying instead on a subtraction logic, i.e. manipulating expectations but keeping the outcomes (i.e. the presented shapes) the same (see reply to

point 2 for more detail on this subtraction strategy). We agree with the reviewer that investigating within-trial dynamics of prediction and prediction error signals in future research will be crucial. This will require methods with millisecond temporal resolution, such as intracranial EEG or MEG. In fact, our lab is currently planning MEG experiments to investigate communication between hippocampus and sensory cortex in a time-resolved manner, probing relationships between hippocampal theta phase and sensory representations during perceptual prediction. We now discuss these limitations and suggestions for future research in the Discussion (p. 14):

“Similarly, the slow nature of the BOLD signal prevents investigating fast within-trial dynamics of prediction and prediction error signals. For instance, it may be that prediction signals always precede prediction error signals in the hippocampus. The hippocampal representations revealed here reflect a temporal integration of neural signals over the course of a trial, and thus indicate whether predictions or prediction errors dominate. Future studies with millisecond temporal resolution are required^{9,82} to reveal the dynamic interplay of predictions and errors within the hippocampus.”

We have also rephrased in several places throughout the manuscript, in the Abstract, Introduction, and Discussion, to avoid giving the impression of a binary flip between modes, but rather a transition in whether errors or predictions dominate in the hippocampus as learning proceeds.

We discuss how our subtraction logic allows us to reveal prediction-error-like and prediction-like signals using fMRI in reply to point 2 below (see also our reply to Reviewer 1 point 4).

2. The strategy of subtracting classifier evidence for Valid – Invalid trials is unclear.

Related to the first major point above, I found other aspects of Fig. 1 difficult to interpret. In particular, the strategy of subtracting classifier evidence for Valid – Invalid trials is not clear; this is important for understanding the results and could be explained more clearly in the methods and figure. I was confused by the correspondence between panels 1E and 1F. It seems that the reader is supposed to look across each color-coded row.

The top row in green indicates that when A is expected and A occurs, we should be able to decode the prediction for A, but not B (because no prediction error occurred). This row makes sense. The middle row in red indicates that when B is expected but A occurs, we should be able to initially decode the prediction for B, but this should switch to A when the unexpected outcome appears. This row is confusing if activation patterns were actually averaged across the whole trial; wouldn't the evidence for A and B cancel out?

The bottom row in yellow is also confusing. On the right, the “predicted shape effect” is defined as the difference between A – B trials during the prediction phase (controlling for the same outcome). On the right, we see the net result for decoder evidence after this subtraction step. I was under the impression that the 1F should now be read from top to bottom: A (Valid, green bar) is subtracted from B (Invalid, red bar) to yield the yellow bar. The “Prediction error” column in 1F aligns with this logic. However, the “Prediction” column does not. Why is the net result (yellow bar) strong evidence in favor of A?

Apologies for not making the subtraction logic sufficiently clear. As our reply to point 1 indicates, the analyses and hypotheses pertain to activity patterns averaged across whole trials. In other words, the hypotheses in Fig. 1f simply pertain to “what if the hippocampus encodes prediction errors” and “what if the hippocampus encodes predictions?”, rather than assuming a sequence of error and prediction signals within trials. We have improved and extended the discussion of the subtraction logic in the manuscript as follows (p. 6):

“Multivoxel decoding analyses (Supplementary Fig. 1), trained on data from separate shape-only runs in which no predictive cues were presented (Fig. 1c, d), were used to reveal hippocampal shape representations on valid and invalid trials (Fig. 1e). If the hippocampus were to represent prediction errors, valid trials should not result in a shape representation, since the predicted and presented shapes are identical and should cancel each other out (Fig. 1f, top left). On invalid trials on the other hand, if shape B is predicted but shape A is presented, unexpected shape A should be represented in the hippocampus (Fig. 1f, middle left). If instead the hippocampus were to represent predictions rather than errors, on invalid trials where shape B is predicted but shape A is presented, shape B should be represented in the hippocampus (Fig. 1f, middle right). Further, on valid trials the shape that is both predicted and presented should be represented (Fig. 1f, top right).

Both of these types of patterns have been observed in the hippocampus⁵⁶, and the aim of the current study was to investigate how they develop over the course of learning. [...]

The clearest way to dissociate effects of the predictive cues from the effects of the presented shapes is to subtract decoding evidence for the invalidly predicted shapes from evidence for the validly predicted shapes (Fig. 1e), since the presented shapes were identical in both types of trials. Under a prediction error hypothesis, this would result in a negative signal (subtracting a positive signal on invalid trials from a zero signal on valid trials; Fig. 1f left column). Under a prediction hypothesis on the other hand this would result in a positive signal (subtracting a negative signal on invalid trials from a positive one on valid trials; Fig. 1f right column). This subtraction therefore constitutes our main effect of interest.”

We have also reformatted Fig. 1e and 1f to depict the hypotheses more clearly, reproduced here for convenience. Specifically, we have drawn boxes around the prediction error and prediction hypotheses, and added minus and equal signs to Fig. 1f, to make it clear that they are not intended to depict subsequent signals within trials, but alternative (static) hypotheses.

Fig. 1e, f. e Subtracting the response evoked by invalidly from validly predicted shapes isolated the effect of the predictive cues. f Hypothesised shape decoding results if the hippocampus represents either prediction errors (left column) or predictions (right column).

3. There are discrepancies between the results for Exp 1 and Exp 2 that are not discussed sufficiently.

It seems that the only difference between Exp 1 and Exp 2 is the number of trials per block (32 trials in Exp 1 vs. 128 trials in Exp 2). The authors state that the allocation of trials to blocks is the only difference between the two experiments. Therefore, we should see the same results from Exp 1 recapitulated in the first ~32 trials of Exp 2. Fig. 1 demonstrates that in Exp 1, the effect of Invalid > Valid classifier evidence (the “prediction error representation”) emerged after trial 22.

However, in the Exp 2 results, it looks like the effect of Invalid > Valid only emerged after trial 40 (Fig. 5). That is more trials than the entire block in Exp 1. This suggests that learning (the “prediction error” representation stage) was slower and took longer in Exp 2. This discrepancy between the two experiments should be discussed.

One possibility is that this discrepancy arises from differences in the sliding window analysis (averaging classifier timecourses across bins of trials). In Exp 1, the bins included 4 trials. In Exp 2, the bins were expanded to 16 trials. The authors state that the bin size was increased because each block was 4x longer in Exp 2 relative to Exp 1. However, it’s not clear why it was necessary to equate the number of bins across experiments. Wouldn’t it be more straightforward to compare results across the two experiments if the bin size was the same, but Exp 2 showed more bins (on a longer x-axis)? This would more clearly show whether the Exp 1 timecourse was replicated in Exp 2, before the Exp 2 results diverged as learning progressed and plateaued.

We agree that we should have discussed this difference in the time at which the negative signals emerge in Experiment 1 and 2 more thoroughly in the manuscript. We also agree that it is likely the result of the differences in sliding window length and smoothing between the two experiments, since as the reviewer says the two experiments were identical other than the number of trials per block. The sliding window and smoothing kernel were four times longer in Experiment 2 than in Experiment 1. This was necessary to achieve a similar signal-to-noise ratio in both experiments; there were four times fewer blocks in Experiment 2, since the blocks were four times longer and the fMRI session could not be made any longer than it already was.

Using a sliding window of four trials would have resulted in only $4 \times 4 = 16$ trials per bin in Experiment 2, of which only 4 (25%) would have been invalid, two per shape. Needless to say, this would have resulted in very noisy fMRI signal estimates, especially for the crucial invalid cue trials, and is therefore not a viable analysis approach. (See also our reply to Reviewer 1 Point 6 for more details on the temporal smoothing implemented here.)

Note that Fig. 2b suggests that the negative predicted shape evidence is still decreasing at the end of the 32 trial blocks, as a result of the valid shape evidence (Fig. 2b) becoming increasingly negative, suggesting that perhaps the peak of the negative signal indeed lies later than at the end of the blocks in Experiment 1.

Speculatively, the delayed negative peak in Experiment 2 may also partly have resulted from averaging together a small positive peak around trial 16 (see Fig. 2, especially subiculum; see also our reply to point 4 below for details on this small positive effect early on in Experiment 1) with the subsequent negative signal. Whereas the sliding window in Experiment 1 may have been short enough to separately resolve these two signals, the longer sliding window in Experiment 2 was not, resulting in these two signals cancelling each other out early in the blocks. Note that this initial positive signal in Experiment 1 was quite weak, and only detectable using post-hoc analyses (see response to point 4 for details), making this explanation very speculative. We nevertheless now include it in the Discussion (p. 15), to acknowledge the discrepancy between the two experiments and suggesting a way future research may be able to shed more light on this:

“It appears that the prediction error-like signals emerged later in Experiment 2 than in Experiment 1 (cf. Fig. 2 and 5). This is likely the result of differences in sliding window length and smoothing between the two experiments, combined with the fact that the prediction error signal may still be building by the end of the blocks in Experiment 1 (Fig. 2b). More speculatively, the later negative peak in Experiment 2 may also partly have resulted from averaging together the small early positive peak discussed above (~trial 16-20) with the subsequent negative signal (~trial 24 onwards) (Supplementary Fig. 3). While the sliding window in Experiment 1 was short enough to separately resolve these two signals, the longer sliding window in Experiment 2 was not, resulting in these two signals cancelling each other out early in the blocks. At present this explanation is highly speculative since the initial positive signal in Experiment 1 was detected using post-hoc analyses and needs to be investigated further, as discussed above.”

4. There appear to be effects that do not necessarily align with the hypotheses and are not discussed. The Exp 1 statistics reported pertain to the Invalid > Valid difference that emerges in the trials at the end of each block. However, Fig. 2 suggests that there is also an effect in the opposite direction (Valid > Invalid) in the middle of each block. This effect is most evident for the Subiculum, but also appears to a lesser extent in the CA1 and whole-hippocampus plots. This opposite effect should be acknowledged in the Results, and statistics should be reported for the earlier portions of the learning blocks. Is there an explanation for why this Valid > Invalid effect might occur during the middle phase of learning? If substantial, unexplained fluctuations in decoding evidence can occur in either direction, it casts doubt on the reliability of the measure (especially because the conclusions depend on interpreting fluctuations over trials).

We thank the reviewer for prompting us to inspect these early positive effects more closely. Initially, we had not explored them further since the early positive bump did not reach significance using our cluster-based permutation tests. However, we have now investigated this further by fitting two sigmoids, rather than one, to the data of Experiment 1, to see whether an initial positive curve followed by a negative one would prove a good fit to the data. As in the analysis of Experiment 2, the first sigmoid was constrained to have a midpoint in the first half of the blocks (here, between trial 1 and 16), while the second had a midpoint in the second half (here, between trial 17 and 32), allowing them to capture potential differences between the first and second half of the blocks. Both sigmoids' amplitudes were free to range between -1 and 1, meaning that this analysis imposed no priors on the signs of the curves.

We found that there was no significantly positive early sigmoid in hippocampus as a whole ($t_{23} = 1.13$, $p = 0.27$), nor in CA1 ($t_{23} = 0.77$, $p = 0.45$) or CA2-3-DG ($t_{23} = 0.66$, $p = 0.52$), but there was in the subiculum ($t_{23} = 3.43$, $p = 0.002$), where the early positive peak was largest. We now report these exploratory results in the manuscript (p. 7), and in a new Supplementary Fig. 3, reproduced here for convenience.

Supplementary Fig. 3. Quantification of hippocampal effects in Experiment 1 with two sigmoids. **A** Double sigmoid learning curve fit to predicted shape decoding in hippocampus and its subfields. Horizontal lines indicate significant clusters. Shaded regions indicate SEM. **B** Amplitude parameters of the two sigmoids making up the fitted curve. Error bars indicate SEM. Dots indicate individual participants. ** $p < 0.01$.

We speculate that this early positive prediction signal in the subiculum might reflect early predictions. It is interesting to note that the prediction effects at the end of the long blocks in Experiment 2, as well as in a previous study with a virtually identical paradigm where participants learned the predictive associations before being scanned (Kok & Turk-Browne 2018), were strongest in the subiculum as well. Of course, calculating a prediction error requires a prediction. Perhaps this initial positive bump reflects early, imprecise prediction signals, which lead to strong prediction errors on subsequent invalid trials. We realise this is very speculative, and future research, for instance using even shorter blocks so that the bins can be made even shorter, should investigate this further. We now mention this in the Discussion (pp. 14-15):

“An exploratory post-hoc analysis of Experiment 1 additionally revealed early prediction-like signals in the subiculum, before the prediction error-dominated signals emerged (Supplementary Fig. 3). This initial positive signal could potentially reflect early, imprecise predictions, which lead to strong prediction errors on subsequent invalid trials. This explanation is currently speculative, especially given the post-hoc nature of the analysis. Future research is needed to investigate the early build-up of prediction signals further, for instance using a paradigm with many blocks with only a few cue repetitions each.”

Minor Points

1. The Introduction does not clearly motivate the main conclusion of the paper.

The introduction states that the hippocampus signals prediction errors (i.e., mismatch signals), but can also represent predictions. This statement is framed as a contradiction that must be resolved, but it is not clear why these two functions would be contradictory. Prediction and error can occur at different timepoints (even within the same trial) without being in conflict. Assuming there is some relevant prior knowledge to guide a prediction, the hippocampus should represent that prediction until the feedback/outcome/stimulus arrives. At that point, it should identify whether the outcome matches or mismatches the expectation.

It would be helpful if the introduction more clearly motivated the key idea that once learning is “complete” the hippocampus should be somewhat resilient to encountering random noise. After extensive prior experience, perhaps prediction errors should not lead us to radically revise our models of the world. If some uncertainty/noise in outcomes is to be expected, prediction error ceases to be a learning signal. This message is clear in the Discussion, but is not laid out clearly in the Introduction.

We have now improved the framing of the problem in the Introduction. We certainly agree that both predictions and prediction errors being present in the hippocampus is not a contradiction, but in fact quite likely. However, most studies of the hippocampus report and discuss only one or the other; the hippocampus as a predictor or the hippocampus as a novelty detector. Our results suggest, as the reviewer states, that it is highly likely that the hippocampus does both, and that it is the balance between the two signals that is affected by learning. Indeed, as the reviewer says, once learning is complete, predictions are stronger/more precise and thus less affected by random noise. In other words, expected prediction errors, i.e. expected uncertainty (Yu & Dayan, 2005), should not lead us to update our model of the world. We have made this more clear in the Introduction (p. 3):

“This raises the question of how the hippocampus balances encoding of new associations with the retrieval of existing ones^{29,30}. One way to achieve this would be to emphasise prediction errors when an environment is novel, since these can serve to update one’s internal model of the world³¹. On the other hand, once an environment (and its statistical regularities) have become familiar, prediction errors

may be downweighted and predictions (i.e., retrieval of existing associations) may dominate. That is, once the statistical regularities of an environment are fully learnt, the hippocampus becomes more resilient to prediction errors caused by random fluctuations (i.e., expected uncertainty), since these are no longer considered model updating ('newsworthy') events."

Reviewer #3

Major points

1. The authors claim that the behavioural data (accuracy/RTs) for the shape-matching task are unaffected by the validity of the auditory cue (since this is not directly task-relevant). However, it seems that a surprising first shape could disrupt performance, at least until the proportion of such surprises is learned (as part of the “environment model”). So rather than averaging over trials, could the authors analyse behavioural data as a function of trial, like they did for the fMRI data?

We thank the reviewer for this suggestion. We have now performed these analyses for both experiments, and they did not reveal any behavioural effects of the predictive cues as a function of trial. This is not unexpected, given that the predictive cues were orthogonal to the shape discrimination task, and that there were no effects of the predictive cues on this same task in a previous study with a virtually identical paradigm in which participants learned the cue contingencies before being scanned (Kok & Turk-Browne 2018). We now report these results in the manuscript (p. 6 and p. 10), and we display these results in a new supplementary Fig. 2, reproduced here for convenience.

Supplementary Fig. 2. Behavioural performance over trials. A Task accuracy on trials with validly predicted (green) and invalidly predicted (red) shapes in Experiment 1 (left panel). Difference in accuracy between valid and invalid trials (yellow, right panel). **B** Task accuracy on trials with validly predicted (green) and invalidly predicted (red) shapes in Experiment 2 (left panel). Difference in accuracy between valid and invalid trials (yellow, right panel). Shaded regions indicate SEM.

2. Where are the horizontal bars indicating cluster-level correction for Experiment 2 (ie Fig. 5), or did differences never survive correction? I appreciate that the authors might prefer inferences based on fitting their sigmoid functions (which is probably more sensitive), but this doesn't allow such precise estimation of when significant decoding happens. This is important, because in Experiment 1, significant decoding of the unexpected shape occurs after about 24 trials, whereas in Experiment 2, it doesn't look like this occurs until 50 trials, i.e. twice as long. Why is this? I could not see an obvious procedural difference between experiments that would explain this. (Better temporal localisation would be achieved by correcting for peak statistic rather than cluster, but this is even less likely to reveal significance.)

We did not perform cluster-based permutation tests in Experiment 2, since this experiment was motivated by a specific hypothesis on the nature of the change of the signal in the hippocampus over trials (Fig. 4). Cluster-based permutation tests are ideal when there is no clear a priori hypothesis on the sign and timing of an effect, but as the reviewer says they are not the most sensitive way of testing specific hypotheses. Also, the hypothesis in Experiment 2 predicted an effect that would go from being negative to being positive, crossing zero. Cluster-based permutation tests in such a case would essentially test for the present of two independent effects (a negative and a positive cluster), rather than one dynamic change. This is the reason that we chose to test our effects by 1) fitting curves, and 2) testing the *derivative* of the hippocampal prediction signal over trials (separately in the first and second half of the blocks). (Note that we do now also perform statistical tests to investigate whether the prediction signal is larger than zero at the end of the blocks, see reply to Reviewer 1 point 3.) We now explain this logic more clearly in the manuscript (p. 22):

“As in Experiment 1, the amplitude parameters were submitted to simple *t*-tests to test whether learning curves significantly deviating from zero. Since Experiment 2 was motivated by a specific hypothesis on the nature of change of the hippocampal signal over trials (Fig. 4) we relied on these tests of the dynamics of the signal, rather than cluster-based permutation tests as in Experiment 1.”

And on p. 9:

“[...] we hypothesised that the hippocampus may switch from representing prediction errors (early in learning) to representing predictions (once learning is complete) as learning progresses (Fig. 4). In order to test this hypothesis, we performed a second fMRI experiment, in which participants (N=24) were exposed to the same cues for longer, and tested for potential switches in dynamics by fitting sigmoid learning curves to the decoding evidence over trials.”

With regard to the prediction error effects occurring later in Experiment 2 than in Experiment 1, this is likely the result of the different sliding window lengths and smoothing used in the two experiments. Essentially, the sliding window was four times longer in Experiment 2, so more temporal smoothing occurred. This was necessary to obtain sufficient signal-to-noise ratio, because Experiment 2 contained four times fewer blocks (since the blocks were four times longer). Specifically, using a sliding window of four trials (as in Experiment 1) would have resulted in only $4 \times 4 = 16$ trials per bin in Experiment 2, of which only 4 (25%) would have been invalid, two per shape. Needless to say, this would have resulted in very noisy fMRI signal estimates, especially for the crucial invalid cue trials, and is therefore not a viable analysis approach. We now acknowledge and discuss this limitation in the manuscript (p. 15):

“It appears that the prediction error-like signals emerged later in Experiment 2 than in Experiment 1 (cf. Fig. 2 and 5). This is likely the result of differences in sliding window length and

smoothing between the two experiments, combined with the fact that the prediction error signal may still be building by the end of the blocks in Experiment 1 (Fig. 2b)."

More speculatively, the delayed negative peak in Experiment 2 may also partly have resulted from averaging together a small positive peak around trial 16 (see Fig. 2, and the new Supplementary Fig. 3, especially subiculum; see also our reply to point 5 below for details on this small positive effect early on in Experiment 1) with the subsequent negative signal. Whereas the sliding window in Experiment 1 may have been short enough to separately resolve these two signals, the longer sliding window in Experiment 2 was not, resulting in these two signals cancelling each other out early in the blocks. Note that this initial positive signal in Experiment 1 was quite weak, and only detectable using post-hoc analyses, making this explanation very speculative. We nevertheless now include it in the Discussion (p. 15), to acknowledge the discrepancy between the two experiments and suggesting a way future research may be able to shed more light on this:

"More speculatively, the later negative peak in Experiment 2 may also partly have resulted from averaging together the small early positive peak discussed above (~trial 16-20) with the subsequent negative signal (~trial 24 onwards) (Supplementary Fig. 3). While the sliding window in Experiment 1 was short enough to separately resolve these two signals, the longer sliding window in Experiment 2 was not, resulting in these two signals cancelling each other out early in the blocks. At present this explanation is highly speculative since the initial positive signal in Experiment 1 was detected using post-hoc analyses and needs to be investigated further, as discussed above."

3. The switch to coding the expected stimulus after about 100 trials in Experiment 2 is certainly very interesting, and reconciles previous findings from the same group (Kok & Turk-Browne), but it is still a novel finding (I think) within a continuous analysis learning procedure, so it would be more convincing if it were replicated at least once, ideally with all analyses pre-registered, eg on OSF.

We certainly agree that it is important that these findings are replicated. We have not performed a direct replication of the switch to coding the expected stimulus after about 100 trials here. The COVID-19 pandemic has severely restricted our ability to perform fMRI scans of groups of participants in a reasonable amount of time, not to mention the backlog this has created on scanner bookings. However, we wish to note that the negative effect of Experiment 1 was replicated in Experiment 2, and the positive effect at the end of the blocks in Experiment 2 is consistent with previous findings (Kok & Turk-Browne 2018), as the reviewer says. Also, the switch was demonstrated by two very different statistical tests, that is by 1) fitting sigmoid curves and 2) testing the derivative of the prediction signal over time separately in the first and second half of the blocks, revealing a positive derivative in the second half of the blocks in Experiment 2.

We do acknowledge that it is still very important to replicate these findings, and we plan to do so by taking this experiment to the 7T MR scanner. This will allow us to 1) replicate these findings and 2) extend them by allowing us to segment the hippocampal subfields in more detail, as well as tests these effects in the different layers of the entorhinal cortex, a major interface between the hippocampus and cortex. We have already started data collection on what is essentially a replication of Kok & Turk-Browne (2018) at 7T, pre-registered on AsPredicted.org ("Direction of communication between cortex and hippocampus", #70936). A replication of the current study is planned to start after the current 7T study is completed.

4. Would it be helpful to have an extra control condition, to control for "time-within-block" (e.g. which

might affect interest/attention to the auditory cues). Such a condition could have no clear auditory-visual contingency for example, so would not be expected to show much decoding.

Investigating the effect of time-within-blocks on attention to the auditory cues is an excellent suggestion. It is certainly possible that, as learning progressed, attention to the auditory cues increased or decreased.

A condition without clear auditory-visual contingencies, as the reviewer suggests, would serve to control for non-specific attention effects. It should be noted however that our analyses were designed to reveal shape-specific activity patterns associated with the auditory cues, and a control condition without any auditory-visual contingencies would therefore not be expected to reveal any effect in our analyses at any time during the blocks. Alternatively, one could design an experiment where the cues differ in their reliability between blocks, e.g., 60-40% in one block and 90-10% in another. This would be expected to affect learning rate, and might speed up the dynamic switch due to learning, but not affect non-specific fluctuations in attention due to time-within-block. Additionally, one could include an explicit manipulation of attention to the auditory cues, e.g. by asking participants to perform a cover task on the auditory cues on half of the blocks. Such an experiment would allow one to dissociate learning effects from fluctuations in alertness/attention levels. We now discuss this idea for future research in the Discussion (p. 15):

“Despite the predictive associations being implicit, hippocampal signals may still have been affected by fluctuations in the level of attention paid to the cues over the course of the blocks. That is, if participants pay more (less) attention to the cues over time, this might increase (decrease) the strength of the prediction signals in the hippocampus. Future research might dissociate learning dynamics and attentional fluctuations by changing the reliability of predictive cues between blocks. More reliable (e.g., 90% valid) cues would be expected to lead to faster learning rates than less reliable (e.g., 60% valid) ones, without affecting non-specific fluctuations in attention due to time spent on task.”

Minor points

5. In Experiment 1, there is an earlier divergence between expected and unexpected stimuli around 16 trials (at least in hippocampus and subiculum), favouring decoding of expecting stimuli. I appreciate that this does not reach significance, and is not obvious in Experiment 2, and therefore likely to be noise, but would it be significant if fit with a second, even earlier sigmoid in Experiment 1? It would be reassuring to know that the sigmoidal fitting is robust to overfitting such potential noise.

We thank the reviewer for prompting us to inspect these early positive effects more closely. Initially, we had not explored them further since the early positive bump did not reach significance using our cluster-based permutation tests. However, we have now investigated this further by fitting two sigmoids, rather than one, to the data of Experiment 1, as the reviewer suggests. As in the analysis of Experiment 2, the first sigmoid was constrained to have a midpoint in the first half of the blocks (here, between trial 1 and 16), while the second had a midpoint in the second half (here, between trial 17 and 32), allowing them to capture potential differences between the first and second half of the blocks. Both sigmoids' amplitudes were free to range between -1 and 1, meaning that this analysis imposed no priors on the signs of the curves.

We found that there was no significantly positive early sigmoid in hippocampus as a whole ($t_{23} = 1.13$, $p = 0.27$), nor in CA1 ($t_{23} = 0.77$, $p = 0.45$) or CA2-3-DG ($t_{23} = 0.66$, $p = 0.52$), but there was in the subiculum

($t_{23} = 3.43$, $p = 0.002$), where the early positive peak was largest. We now report these exploratory results in the manuscript (p. 7), and in a new supplementary Fig. 3, reproduced on p. 15 of this document for convenience.

We speculate that this early positive prediction signal in the subiculum might reflect early predictions. It is interesting to note that the prediction effects at the end of the long blocks in Experiment 2, as well as in a previous study with a virtually identical paradigm where participants learned the predictive associations before being scanned (Kok & Turk-Browne 2018), were strongest in the subiculum as well. Of course, calculating a prediction error requires a prediction. Perhaps this initial positive bump reflects early, imprecise prediction signals, which lead to strong prediction errors on subsequent invalid trials. We realise this is very speculative, and future research, for instance using even shorter blocks so that the bins can be made even shorter, should investigate this further. We now mention this in the Discussion (p. 14-15):

“An exploratory post-hoc analysis of Experiment 1 additionally revealed early prediction-like signals in the subiculum, before the prediction error-dominated signals emerged (Supplementary Fig. 3). This initial positive signal could potentially reflect early, imprecise predictions, which lead to strong prediction errors on subsequent invalid trials. This explanation is currently speculative, especially given the post-hoc nature of the analysis. Future research is needed to investigate the early build-up of prediction signals further, for instance using a paradigm with many blocks with only a few cue repetitions each.”

6. Is correction for multiple comparisons needed across the number of ROIs (subfields)?

The hierarchical logic behind our analyses was to test the effects in the hippocampus as a whole first, and following up any significant effects by investigating potential differences between the subfields (and anterior and posterior hippocampus). In other words, the tests of the different ROIs were not multiple independent simultaneous tests, but sequential ones following a hierarchical approach, with tests of subdivisions of ROIs following up significant effects in the ROI as a whole. For instance, that is why we split the subfields for posterior hippocampus in Experiment 2 but not in Experiment 1; because there was a significant anterior vs. posterior effect in Experiment 2 but not Experiment 1. This hierarchical approach helps control the false positive rate, rather than simply examining all possible comparisons. We have now made this logic clearer in the manuscript (p. 22):

“All analyses were initially performed on the hippocampus ROI as a whole, and when significant these were followed up by investigating hippocampal subfields and comparing the anterior and posterior hippocampi. This hierarchical approach, where significant effects in the ROI as a whole were followed up with tests of its subdivisions, rather than simply examining all possible comparisons, helped control the false positive rate.”

7. Extraneous “bar” after ref 30 on line 59

This has been corrected.

Thank you for providing the data and code. I did not have time to run it all, but it did help me confirm aspects of the analysis like the randomisation used for cluster-level correction.

REVIEWERS' COMMENTS

Reviewer #1 (Remarks to the Author):

The authors have done a commendable job revising the manuscript and addressing reviewer comments. I have reviewed all response to comments that I raised as well as comments from the other reviewers. Overall, I am satisfied with the revisions and I believe the manuscript has been strengthened. It is notable that all three reviewers commented on the apparently different timing of effects across E1 and E2. The primary explanation for this offered by the authors seems to be that it is an artifact of different smoothing windows. It seems that this argument could potentially be more directly supported by showing (even if just for qualitative purposes) unsmoothed data. That said, I think the key point (which the authors have addressed) is that this apparent discrepancy is at least now acknowledged in the manuscript and speculative interpretations are provided with appropriate caveats. Otherwise, I found the responses to be helpful in terms of clarifying and/or strengthening the arguments. I believe this will be a paper of broad interest to readers.

Reviewer #2 (Remarks to the Author):

The authors have done an excellent job addressing the comments posed in the reviews; I have no further comments.

Reviewer #3 (Remarks to the Author):

I am happy enough with the authors' responses to all three reviews.

We thank the reviewers for their positive assessment of our revised manuscript. Our response to the remaining minor point is detailed below.

Reviewer #1

1. The authors have done a commendable job revising the manuscript and addressing reviewer comments. I have reviewed all response to comments that I raised as well as comments from the other reviewers. Overall, I am satisfied with the revisions and I believe the manuscript has been strengthened. It is notable that all three reviewers commented on the apparently different timing of effects across E1 and E2. The primary explanation for this offered by the authors seems to be that it is an artifact of different smoothing windows. It seems that this argument could potentially be more directly supported by showing (even if just for qualitative purposes) unsmoothed data. That said, I think the key point (which the authors have addressed) is that this apparent discrepancy is at least now acknowledged in the manuscript and speculative interpretations are provided with appropriate caveats. Otherwise, I found the responses to be helpful in terms of clarifying and/or strengthening the arguments. I believe this will be a paper of broad interest to readers.

We thank the reviewer for this positive feedback. We agree that performing the analysis on unsmoothed data would be ideal, but as we explained in our previous response this was not feasible. The fMRI response to individual trials is relatively noisy, and therefore averaging multiple trials together is necessary to obtain sufficient signal-to-noise ratio. Therefore we performed the time-resolved analyses in the current study using a sliding window approach, followed by temporal smoothing. Experiment 2 contained four blocks of 128 trials (rather than 16 blocks of 32 trials in Experiment 1), and therefore required a longer sliding window and larger smoothing kernel to obtain sufficient signal-to-noise ratio. In the previous revision, we did perform a control analysis I which we did not apply explicit smoothing to the data, which confirmed that our results did not depend on the exact sliding window and smoothing kernel (reproduced below in Figure R1).

Fig. R1. Shape decoding over trials in both experiments, longer sliding window but no smoothing afterwards. Left panels: Decoding evidence for validly (green) and invalidly (red) predicted shapes in the hippocampus in

Experiment 1 (top) and Experiment 2 (bottom). Middle panels: Decoding evidence for predicted (valid – invalid) shapes in hippocampus in Experiment 1 (top) and Experiment 2 (bottom). In Experiment 1, cluster-based permutation tests reveal significant evidence for invalidly predicted shapes ($p = 0.041$) and significantly

However, this analysis still used a sliding window approach, meaning that some smoothing still occurs. A control analysis without any smoothing is not feasible on the current data. This is a limitation of the study that might be addressed by collecting much larger amounts of data, e.g. conducting four or more fMRI sessions per participant using the paradigm of Experiment 2. This is beyond the scope of the current study, but we now mention this limitation and potential resolution in the manuscript (p. 21):

“In the current study, analysing time courses without applying either a sliding window or temporal smoothing was not feasible, as fMRI responses to individual trials are not sufficiently robust. Future work could potentially address this by conducting multiple (e.g., four or more) fMRI sessions per participant, increasing the amount of data per trial position.”

Reviewer #2

The authors have done an excellent job addressing the comments posed in the reviews; I have no further comments.

Reviewer #3

I am happy enough with the authors' responses to all three reviews.